# Smooth Bilevel Programming
# for Sparse Regularization

**Clarice Poon,**[*]    **Gabriel Peyré**[†]

## Abstract

Iteratively reweighted least square (IRLS) is a popular approach to solve sparsity-enforcing regression problems in machine learning. State of the art approaches are more efficient but typically rely on specific coordinate pruning schemes. In this work, we show how a surprisingly simple re-parametrization of IRLS, coupled with a bilevel resolution (instead of an alternating scheme) is able to achieve top performances on a wide range of sparsity (such as Lasso, group Lasso and trace norm regularizations), regularization strength (including hard constraints), and design matrices (ranging from correlated designs to differential operators). Similarly to IRLS, our method only involves linear systems resolutions, but in sharp contrast, corresponds to the minimization of a smooth function. Despite being non-convex, we show that there are no spurious minima and that saddle points are "ridable", so that there always exists a descent direction. We thus advocate for the use of a BFGS quasi-Newton solver, which makes our approach simple, robust and efficient. We perform a numerical benchmark of the convergence speed of our algorithm against state of the art solvers for Lasso, group Lasso, trace norm and linearly constrained problems. These results highlight the versatility of our approach, removing the need to use different solvers depending on the specificity of the ML problem under study.

## 1 Introduction

Regularized empirical risk minimization is a workhorse of supervised learning, and for a linear model, it reads

$$\min_{\beta \in \mathbb{R}^n} R(\beta) + \frac{1}{\lambda} L(X\beta, y) \tag{$\mathcal{P}_\lambda$}$$

where $X \in \mathbb{R}^{m \times n}$ is the design matrix ($n$ being the number of samples and $m$ the number of features), $L : \mathbb{R}^m \times \mathbb{R}^m \to [0, \infty)$ is the loss function, and $R : \mathbb{R}^n \to [0, \infty)$ the regularizer. Here $\lambda \geqslant 0$ is the regularisation parameter which is typically tuned by cross-validation, and in the limit case $\lambda = 0$, $(\mathcal{P}_0)$ is a constraint problem $\min_\beta R(\beta)$ under the constraint $L(X\beta, y) = 0$.

In this work, we focus our attention to sparsity enforcing penalties, which induce some form of structure on the solution of $(\mathcal{P}_\lambda)$, the most celebrated examples (reviewed in Section 2) being the Lasso, group-Lasso and trace norm regularizers. All these regularizers, and much more (as detailed in Section), can be conveniently re-written as an infimum of quadratic functions. While Section 2 reviews more general formulations, this so-called "quadratic variational form" is especially simple in the case of block-separable functionals (such as Lasso and group-Lasso), where one has

$$R(\beta) = \min_{\eta \in \mathbb{R}^k_+} \frac{1}{2} \sum_{g \in \mathcal{G}} \frac{\|\beta_g\|_2^2}{\eta_g} + \frac{1}{2} h(\eta), \tag{1}$$

---

[*]Department of mathematical sciences, University of Bath, Bath BA2 7AY, UK `cmshp20@bath.ac.uk`

[†]CNRS and DMA, Ecole Normale Supérieure, PSL University, 45 rue d'Ulm, F-75230 PARIS cedex 05, FRANCE, `gabriel.peyre@ens.fr`

35th Conference on Neural Information Processing Systems (NeurIPS 2021).

where $\mathcal{G}$ is a partition of $\{1, \ldots, n\}$, $k = |\mathcal{G}|$ is the number of groups and $h : \mathbb{R}_+^k \to [0, \infty)$. An important example is the group-Lasso, where $R(\beta) = \sum_g \|\beta_g\|_2$ is a group-$\ell_1$ norm, in which case $h(\eta) = \sum_i \eta_i$. The special case of the Lasso, corresponding to the $\ell_1$ norm is obtained when $g = \{i\}$ for $i = 1, \ldots, n$ and $k = n$. This quadratic variational form (1) is at the heart of the Iterative Reweighted Least Squares (IRLS) approach, reviewed in Section 1.1. We refer to Section 2 for an in-depth exposition of these formulations.

Sparsity regularized problems $(\mathcal{P}_\lambda)$ are notoriously difficult to solve, especially for small $\lambda$, because $R$ is a non-smooth function. It is the non-smoothness of $R$ which forces the solutions of $(\mathcal{P}_\lambda)$ to belong to low-dimensional spaces (or more generally manifolds), the canonical example being spaces of sparse vectors when solving a Lasso problem. We refer to Bach et al. [2011] for an overview of sparsity-enforcing regularization methods. The core idea of our algorithm is that a simple re-parameterization of (1) combined with a bi-level programming (i.e. solving two nested optimization problems) can turn $(\mathcal{P}_\lambda)$ into a smooth program which is much better conditioned, and can be tackled using standard but highly efficient optimization techniques such as quasi-Newton (L-BFGS). Indeed, by doing the change of variable $(v_g, u_g) \triangleq (\sqrt{\eta_g}, \beta_g/\sqrt{\eta_g})$ in (1), $(\mathcal{P}_\lambda)$ is equivalent to

$$\min_{v \in \mathbb{R}^k} f(v) \quad \text{where} \quad f(v) \triangleq \min_{u \in \mathbb{R}^n} G(u, v) \tag{2}$$

$$G(u, v) \triangleq \frac{1}{2} h(v \odot v) + \frac{1}{2}\|u\|^2 + \frac{1}{\lambda} L(X(v \odot_{\mathcal{G}} u), y). \tag{3}$$

Throughout, we define $\odot$ to be the standard Hadamard product and for $v \in \mathbb{R}^k$ and $u \in \mathbb{R}^n$, we define $v \odot_{\mathcal{G}} u \in \mathbb{R}^n$ to be such that $(v \odot_{\mathcal{G}} u)_g = v_g u_g$. Provided that $v \mapsto h(v \odot v)$ is differentiable and $L(\cdot, y)$ is a convex, proper, lower semicontinuous function, the inner minimisation problem has a unique solution and $f$ is differentiable. Moreover, in the case of the quadratic loss $L(z, y) \triangleq \frac{1}{2}\|z - y\|_2^2$, the gradient of $f$ can be computed in closed form, by solving a linear system of dimension $m$ or $n$. This paper is thus devoted to study the theoretical and algorithmic implications of this simple twist on the celebrated IRLS approach.

**Comparison with proximal gradient** To provide some intuition about the proposed approach, the figure on the right contrasts the iterations of a gradient descent on $f$ and of the iterative soft thresholding algorithm (ISTA) on the Lasso. We consider a random Gaussian matrix $X \in \mathbb{R}^{10 \times 20}$ with $\lambda = \|X^\top y\|_\infty/10$. The ISTA trajectory is non-smooth when some feature crosses 0. In particular, if a coefficient (such as the red one on the figure) is initialized with the wrong sign, it takes many iteration for ISTA to flip sign. In sharp contrast, the gradient flow of $f$ does not exhibit such a singularity and exhibits a smooth geometric convergence. We refer to the appendix for an analysis of this phenomenon.

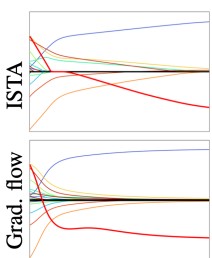

**Contributions** Our main contribution is a new versatile algorithm for sparse regularization, which applies standard smooth optimization methods to minimize the function $f$ in (2). We first propose in Section 2 a generic class of regularizers $R$ that enjoy a quadratic variational form. This section recaps existing results under a common umbrella and shows the generality of our approach. Section 3.1 then gathers the theoretical analysis of the method, and in particular the proof that while being non-convex, the function $f$ has no local minimum and only "ridable" saddle points. As a result, one can guarantee convergence to a global minimum for many optimisation schemes, such as gradient descent with random perturbations [Lee et al., 2017, Jin et al., 2017] or trust region type methods [Pascanu et al., 2014]. Furthermore, for the case of the group Lasso, we show that $f$ is an infinitely differentiable function with uniformly bounded Hessian. Consequently, standard solvers such as Newton's method/BFGS can be applied and with a superlinear convergence guarantee. Section 4 performs a detailed numerical study of the method and benchmarks it against several popular competing algorithms for Lasso, group-Lasso and trace norm regularization. Our method is consistently amongst the best performers, and is in particular very efficient for small values of $\lambda$, and can even cope with the constrained case $\lambda = 0$.

## 1.1 Related works

**State of the art solvers for sparse optimisation** Popular approaches to nonsmooth sparse optimisation include proximal based methods. The simplest instance is the Forward-Backward algorithm

[Lions and Mercier, 1979, Daubechies et al., 2004] which handles the case where $L$ is a smooth function. There are many related inertial based acceleration schemes, such as FISTA [Beck and Teboulle, 2009] and particularly effective techniques that leads to substantial speedups are the adaptive use of stepsizes and restarting strategies [O'donoghue and Candes, 2015]. Other related approaches are proximal quasi-Newton and variable metric methods [Combettes and Vũ, 2014, Becker et al., 2019], which incorporate variable metrics into the quadratic term of the proximal operator. Another popular approach, particularly for the Lasso-like problems are coordinate descent schemes [Friedman et al., 2010]. These schemes are typically combined with support pruning schemes [Ghaoui et al., 2010, Ndiaye et al., 2017, Massias et al., 2018]. In the case where both the regularisation and loss terms are nonsmooth (e.g. in the basis pursuit setting), typical solvers are the primal-dual [Chambolle and Pock, 2011], ADMM [Boyd et al., 2011] and Douglas-Rachford algorithms [Douglas and Rachford, 1956]. Although these schemes are very popular due to their relatively low per iteration complexity, these methods have sublinear convergence rates in general, with linear convergence under strong convexity.

**The quadratic variational formulation and IRLS**   Quadratic variational formulations such as (1) have been exploited in many early computer vision works, such as Geiger and Yuille [1991] and Geman and Reynolds [1992]. One of the first theoretical results can be found in Geman and Reynolds [1992], see also Black and Rangarajan [1996] which lists many examples. Our result in Theorem 1 can be seen as a generalisation of these early works. Norm regularizers of this form were introduced in Micchelli et al. [2013], and further studied in the monograph Bach et al. [2011] under the name of subquadratic norms. The main algorithmic consequence of the quadratic variational formulation in the literature is the iterative reweighted least squares (IRLS) algorithm. When $L(z, y) = \frac{1}{2}\|z - y\|_2^2$ is the quadratic loss, a natural optimisation strategy is alternating minimising. However, due to the $1/\eta_g$ term, one needs to introduce some regularisation to ensure convergence. One popular approach is to add $\frac{\varepsilon}{2} \sum_g \eta_g^{-1}$ to the formulation (1) which leads to the IRLS algorithm [Daubechies et al., 2010]. The $\varepsilon$-term can be seen as a barrier to keep $\eta_g$ positive which is reminiscent of interior point methods. The idea of IRLS is to do alternating minimisation over $\beta$ and $\eta$, where the minimisation with respect to $\beta$ is a least squares problem, and the minimisation with respect to $\eta$ admits a closed form solution. A nuclear norm version of IRLS has been used in Argyriou et al. [2008] where an alternating minimisation algorithm was introduced. Finally, we remark that although nonconvex formulations for various low complexity regularizers have appeared in the literature, see for instance Rennie and Srebro [2005], Hastie et al. [2015], Mardani and Giannakis [2015], Hoff [2017] for the case of the $\ell_1$ and nuclear norms, they are typically associated with alternating minimisation algorithms.

**Variable projection/reduced gradient approaches**   IRLS methods are quite slow because the resulting minimization problem is poorly conditioned. Adding the smoothing term $\frac{\varepsilon}{2} \sum_g \eta_g^{-1}$ only partly alleviates this, and also breaks the sparsity enforcing property of the regularizer $R$. We avoid both issues in (2) by solving a "reduced" problem which is much better conditioned and smooth. This idea of solving a bi-variate problem by re-casting it as a bilevel program is classical, we refer in particular to [Rockafellar and Wets, 2009, Chap. 10] for some general theoretical results on reduced gradients, and also Danskin's theorem (although this is restricted to the convex setting) in Bertsekas [1997]. Our formulation falls directly into the framework of variable projection [Ruhe and Wedin, 1980, Golub and Pereyra, 2003], introduced initially for solving nonlinear least squares problems. Properties and advantages of variable projection have been studied in Ruhe and Wedin [1980], we refer also to Hong et al. [2017], Zach and Bourmaud [2018] for more recent studies. Nonsmooth variable projection is studied in van Leeuwen and Aravkin [2016], although the present work is in the classical setting of variable projection due to our smooth reparametrization. Reduced gradients have also been associated with the quadratic variational formulation in several works [Bach et al., 2011, Pong et al., 2010, Rakotomamonjy et al., 2008]. The idea is to apply descent methods over $g(\eta) = \min_\beta R_0(\eta, \beta) + \frac{1}{2}\|X\beta - y\|_2^2$. Although the function over $\eta$ and $\beta$ is discontinuous, the function $g$ over $\eta$ is smooth and one can apply first order methods, such as proximal gradient descent to minimise $g$ under positivity constraints. While quasi-Newton methods can be applied in this setting with bound constraints, we show in Section 4.1 that this approach is typically less effective than our nonconvex bilevel approach. In the setting of the trace norm, the optimisation problem is constrained on the set of positive semidefinite matrices, so one is restricted to using first order methods [Pong et al., 2010].

## 2 Quadratic variational formulations

We describe in this section some general results about when a regulariser has a quadratic variational form. Our first result brings together results which are scattered in the literature: it is closely related to Theorem 1 in Geman and Reynolds [1992], but their proof was only for strictly concave differentiable functions and did not explicitly connect to convex conjugates, while the setting for norms have been characterized in the monograph Bach et al. [2011] under the name of subquadratic norms.

**Theorem 1.** *Let $R : \mathbb{R}^n \to \mathbb{R}$. The following are equivalent:*
*(i) $R(\beta) = \varphi(\beta \odot \beta)$ where $\varphi$ is proper, concave and upper semi-continuous, with domain $\mathbb{R}_+^d$.*
*(ii) There exists a convex function $\psi$ for which $R(\beta) = \inf_{z \in \mathbb{R}_+^n} \frac{1}{2} \sum_{i=1}^n z_i \beta_i^2 + \psi(z)$.*
*Furthermore, $\psi(z) = (-\varphi)^*(-z/2)$ is defined via the convex conjugate $(-\varphi)^*$ of $-\varphi$, leading to* (1) *using the change of variable $\eta \leftarrow 1/z$ and $h(\eta) = 2\psi(1/\eta)$. When $R$ is a norm, the function $h$ can be written in terms of the dual norm $R^*$ as $h(\eta) = \max_{R^*(w) \leqslant 1} \sum_i w_i^2 \eta_i$. Moreover, $R(\beta)^2 = \inf_{\eta \in \mathbb{R}_+^n} \left\{ \sum_i \beta_i^2 \eta_i^{-1} \setminus h(\eta) \leqslant 1 \right\}$.*

See Appendix B for the proof to this Theorem. Some additional properties of $\psi$ are derived in Lemma 1 of Appendix B, one property is that if $R$ is coercive, then $\lim_{\|z\| \to 0} \psi(z) = +\infty$, so the function $f$ is coercive (see also remark 2).

### 2.1 Examples

Let us first give some simple examples of both convex and non-convex norms:
- *Euclidean norms:* for $R = \| \cdot \|_2$, making use of $R^*$ as stated in Theorem 1, one has $h = \| \cdot \|_\infty$.
- *Group norms:* for $\mathcal{G}$ is a partition of $\{1, \ldots, n\}$, the group norm is $R(\beta) = \sum_{g \in \mathcal{G}} \|\beta_g\|$. Using the previous result for the Euclidean norm, one has $h(\eta) = \sum_{g \in \mathcal{G}} \|(\eta_i)_{i \in g}\|_\infty$. This expression can be further simplified to obtain (1) with a reduced vector $\eta$ in $\mathbb{R}_+^{|\mathcal{G}|}$ in place of $\mathbb{R}^n$ by noticing that the optimal $\eta$ is constant in each group $g$.
- *$\ell^q$ (quasi) norms:* For $R(\beta) = |\beta|^q$ where $q \in (0, 2)$, one has $\varphi(u) = u^{q/2}$ and one verifies that $h(\eta) = C_q \eta^{\frac{q}{2-q}}$ where $C_q = (2 - q) q^{q/(2-q)}$. Note that for $q > \frac{2}{3}$, $v \mapsto h(v^2) = v^\gamma$ for $\gamma > 1$ is differentiable. Analysis and numerics for this nonconvex setting can be found in Appendix F.

**Matrix regularizer** The extension of Theorem 1 to the case where $\beta = B$ is a matrix can be found in the appendix. When $R = \varphi(BB^\top)$ is a function on matrices, the analogous quadratic variational formulation is

$$R(B) = \min_{Z \in \mathbb{S}_+^n} \min_{B \in \mathbb{R}^{n \times r}} \frac{1}{2} \sum_g \operatorname{tr}(B^\top Z^{-1} B) + \frac{1}{2} h(Z), \tag{4}$$

where $\mathbb{S}_+^n$ denotes the set of symmetric positive semidefinite matrices and $h(Z) = 2(-\varphi)^*(-Z^{-1}/2)$. Letting $U = Z^{-1/2} B$ and $V = Z^{1/2}$, we have $B = VU$ and the equivalence

$$\min_B R(B) + \frac{1}{2\lambda} \|\mathcal{A}(B) - y\|_2^2 \tag{5}$$

$$= \min_{V \in \mathbb{R}^{n \times n}} f(V) \triangleq \min_{U \in \mathbb{R}^{n \times r}} \frac{1}{2} \|U\|_F^2 + \frac{1}{2} h(V^\top V) + \frac{1}{2\lambda} \|\mathcal{A}(VU) - y\|_2^2. \tag{6}$$

where $\mathcal{A} : \mathbb{R}^{n \times r} \to \mathbb{R}^m$ is a linear operator. Again, provided that $V \mapsto h(V^\top V)$ is differentiable, $f$ is a differentiable function with $\nabla f(V) = V \partial h(V^\top V) + \frac{1}{\lambda} \mathcal{A}^*(\mathcal{A}(VU) - y)) U^\top$ and $U$ such that $\lambda U + V^\top \mathcal{A}^*(\mathcal{A}(VU) - y) = 0$. For the trace norm, $R(B) = \operatorname{tr}(\sqrt{B^\top B})$, we have $h(Z) = \operatorname{tr}(Z)$ and $\partial h(Z) = \mathrm{Id}$. Note that, just in the vectorial case, one could write the inner minimisation problem over the dual variable $\alpha \in \mathbb{R}^m$ and handle the case of $\lambda = 0$.

## 3 Theoretical analysis

Our first result shows the equivalence between $(\mathcal{P}_\lambda)$ and a smooth bilevel problem.

**Theorem 2.** *Denote $L_y \triangleq L(\cdot, y)$ and let $L_y^*$ denote the convex conjugate of $L_y$. Assume that $L_y$ is a convex, proper, lower semicontinuous function and $R$ takes the form* (1). *The problem* $(\mathcal{P}_\lambda)$ *is equivalent to*

$$\min_{v \in \mathbb{R}^k} f(v) \triangleq \min_{u \in \mathbb{R}^n} \frac{1}{2} h(v \odot v) + \frac{1}{2}\|u\|^2 + \frac{1}{\lambda} L_y\left(X(v \odot_{\mathcal{G}} u)\right). \tag{7}$$

$$= \max_{\alpha \in \mathbb{R}^m} \frac{1}{2} h(v \odot v) - \frac{1}{\lambda} L_y^*(\lambda\alpha) - \frac{1}{2}\|v \odot_{\mathcal{G}} (X^\top \alpha)\|_2^2. \tag{8}$$

*where $v \odot_{\mathcal{G}} u \triangleq (u_g v_g)_{g \in \mathcal{G}}$. The minimiser $\beta$ to $(\mathcal{P}_\lambda)$ and the minimiser $v$ to* (7) *are related by $\beta = v \odot_{\mathcal{G}} u = -v^2 \odot_{\mathcal{G}} X^\top \alpha$. Provided that $v \mapsto h(v \odot v)$ is differentiable, the function $f$ is differentiable with gradient*

$$\nabla f(v) = v \odot \partial h(v^2) - v \odot_{\mathcal{G}} \left(\|X_g^\top \alpha\|^2\right)_g \quad \text{where} \quad \alpha \in \operatorname{argmax}_{\tilde{\alpha}} -L_y^*(\tilde{\alpha}) - \frac{1}{2}\|v \odot_{\mathcal{G}}(X^\top \tilde{\alpha})\|_2^2. \tag{9}$$

*Note that $\nabla f$ is uniquely defined even if $\alpha$ is not unique. If $\lambda > 0$ and $L_y$ is differentiable, then we have the additional formula, with $u \in \operatorname{argmin}_{\tilde{u}} \frac{1}{2}\|\tilde{u}\|_2^2 + \frac{1}{\lambda} L_y(X(v \odot_{\mathcal{G}} \tilde{u}))$,*

$$\nabla f(v) = v \odot \partial h(v^2) + \frac{1}{\lambda}\left(\langle u_g, X_g^\top \partial L_y(X(v \odot_{\mathcal{G}} u))\rangle\right)_{g \in \mathcal{G}}. \tag{10}$$

*Proof.* The equivalence between $(\mathcal{P}_\lambda)$ and (7) is simply due to the quadratic variational form of $R$, and the change of variable $v_g = \sqrt{\eta_g}$, and $u_g = \beta_g/\sqrt{\eta_g}$. The equivalence to (8) follows by convex duality on the inner minimisation problem, that is

$$f(v) = \min_u G_1(v, u) \triangleq \frac{1}{2} h(v^2) + \frac{1}{2}\|u\|^2 + \frac{1}{\lambda} L_y(X(v \odot_{\mathcal{G}} u))$$

$$= \min_{u,z} \frac{1}{2} h(v^2) + \frac{1}{2}\|u\|^2 + \frac{1}{\lambda} L_y(z) \quad \text{where} \quad z = X(v \odot_{\mathcal{G}} u)$$

$$= \min_{u,z} \max_\alpha \frac{1}{2} h(v^2) + \frac{1}{2}\|u\|^2 + \frac{1}{\lambda} L_y(z) - \langle \alpha, z \rangle + \langle \alpha, X(v \odot_{\mathcal{G}} u)\rangle$$

$$= \max_\alpha \min_u \frac{1}{2} h(v^2) + \frac{1}{2}\|u\|^2 + \langle \alpha, X(v \odot_{\mathcal{G}} u)\rangle - \frac{1}{\lambda} L_y^*(\lambda\alpha).$$

Using the optimality condition over $u$, we obtain $u = -v \odot_{\mathcal{G}} X^\top \alpha$ and hence,

$$f(v) = \max_\alpha G_2(v, \alpha) \triangleq \frac{1}{2} h(v^2) - \frac{1}{2}\|v \odot_{\mathcal{G}} X^\top \alpha\|_2^2 - \frac{1}{\lambda} L_y^*(\lambda\alpha).$$

By [Rockafellar and Wets, 2009, Theorem 10.58], if the following set

$$\mathcal{S}(v) \triangleq \left\{\partial_v G_2(v, \alpha) = v - v \odot_{\mathcal{G}} (\|X_g^\top \alpha\|_2^2)_{g \in \mathcal{G}} \setminus \alpha \in \operatorname{argmin}_\alpha G_2(v, \alpha)\right\}$$

is singled valued, then $f$ is differentiable with $\nabla f(v) = \partial_v G_2(v, \alpha)$ for $\alpha \in \operatorname{argmin}_\alpha G_2(v, \alpha)$. Observe that even if $\operatorname{argmin}_\alpha G_2(v, \alpha)$ is not single valued, since $G_2$ is strongly convex for $v \odot_{\mathcal{G}} X^\top \alpha$, $\mathcal{S}(v)$ is single-valued and hence, $f$ is a differentiable function.

In the case where $L_y$ is differentiable, we can again apply [Rockafellar and Wets, 2009, Theorem 10.58], to obtain $\nabla f(v) = v \odot \partial h(v^2) + \partial_v G_1(v, y)$ with $u = \operatorname{argmin}_u G_1(v, u)$ (noting that $G_1(v, \cdot)$ is strongly convex and has a unique minimiser) which is precisely the gradient formula (10). $\qquad\square$

For the Lasso and basis pursuit setting, the gradient of $f$ can be computed in closed form:

**Corollary 1.** *If $R$ has a quadratic variational form and $L_y(z) = \frac{1}{2}\|z - y\|_2^2$, then $\partial L_y(z) = z - y$, $L_y^*(\alpha) = \langle y, \alpha \rangle + \frac{1}{2}\|\alpha\|_2^2$ and the gradient of $f$ can be written as in* (9) *and additionally* (10) *when $\lambda > 0$. Furthermore, $\alpha \in \mathbb{R}^m$ in* (9) *and $u \in \mathbb{R}^n$ in* (10) *solves*

$$(X \operatorname{diag}(\bar{v}^2) X^\top + \lambda \operatorname{Id})\alpha = -y, \quad \text{and} \quad (\operatorname{diag}(\bar{v}) X^\top X \operatorname{diag}(\bar{v}) + \lambda \operatorname{Id})u = v \odot_{\mathcal{G}} (X^\top y), \tag{11}$$

*where $\bar{v} \in \mathbb{R}^n$ is defined as $\bar{v} \odot u = (v_g u_g)_{g \in \mathcal{G}}$ for all $u \in \mathbb{R}^n$ and $\operatorname{Id}$ denotes the identity matrix.*

This shows that our method caters for the case $\lambda = 0$ with the same algorithm in a seamless manner. This is unlike most existing approach which work well for $\lambda > 0$ (and typically do not require matrix inversion) but fails when $\lambda$ is small, whereas solvers dedicated for $\lambda = 0$ might require inverting a linear system, see Section 4.4 for an illustrative example.

## 3.1 Properties of the projected function $f$

In this section, we analyse the case of the group Lasso. The following theorem ensures that the projected function $f$ has only strict saddle points or global minima. We say that $v$ is a second order stationary point if $\nabla f(v) = 0$ and $\nabla^2 f(v) \succeq 0$. We say that $v$ is a strict saddle point (often called "ridable") if it is a stationary point but not a second order stationary point. One can thus always find a direction of descent outside the set of global minimum. This can be exploited to derive convergence guarantees to second order stationary points for trust region methods [Pascanu et al., 2014] and gradient descent methods [Lee et al., 2017, Jin et al., 2017].

**Theorem 3.** *In the case $h(z) = \sum_i z_i$ and $L(z, y) = \frac{1}{2}\|z - y\|_2^2$, the projected function $f$ is infinitely continuously differentiable and for $v \in \mathbb{R}^k$, $\nabla f(v) = v \odot (1 - |\xi|^2)$ where $\xi_g = \frac{1}{\lambda} X_g^\top (X(u \odot v) - y)$ and $u$ solves the inner least squares problem for $v$. Let $J$ denote the support of $v$, by rearranging the columns and rows, the Hessian of $f$ can be written as the following block diagonal matrix*

$$\nabla^2 f(v) = \begin{pmatrix} \operatorname{diag}(1 - \|\xi_g\|_2^2)_{g \in J} + 4U^\top W U & 0 \\ 0 & \operatorname{diag}(1 - \|\xi_g\|_2^2)_{g \in J^c} \end{pmatrix} \tag{12}$$

*where $W \triangleq \operatorname{Id} - \lambda \left((v_g X_g^\top X_h v_h)_{g,h \in J} + \lambda \operatorname{Id}_J\right)^{-1}$ and $U$ is the block diagonal matrix with blocks $(\xi_g)_{g \in J}$, with $\max_{g \in \mathcal{G}} \|\xi_g\|_2 \leqslant C$ and $\|\nabla^2 f(v)\| \leqslant 1 + 3C^2$ where $C \triangleq \|y\|_2 \max_{g \in \mathcal{G}} \|X_g\|/\lambda$. Moreover, all stationary points of $f$ are either global minima or strict saddles. At stationary points, the eigenvalues of the Hessian of $f$ are at most $4$ and is at least*

$$\min\left(4(1 - \lambda/(\lambda + \hat\sigma)), \min_{g \notin J}(1 - \|\xi_g\|^2)\right)$$

*where $\hat\sigma$ is the smallest eigenvalue of $(v_g X_g^\top X_h v_h)_{g,h \in J}$.*

The proof can be found in Appendix C. We simply mention here that by examining the first order condition of $(\mathcal{P}_\lambda)$, we see that $\beta$ is a minimizer if and only if $\xi$ satisfies $-\xi_g = \frac{\beta_g}{\|\beta_g\|_2}$ for all $g \in \operatorname{Supp}(\beta)$ and $\|\xi_g\|_2 \leqslant 1$ for all $g \in \mathcal{G}$. The first condition on the support of $\beta$ is always satisfied at stationary points of the nonconvex function (2), and by examining (12), the second condition is also satisfied unless the stationary point is strict.

*Remark* 1 (Example of strict saddle point for our $f$). One can observe that $v = 0$ is a strict saddle point, as the solution to the associated linear system yields $u = 0$ and hence $\nabla f(v) = 0$. If $\lambda \geqslant \|X^\top y\|_\infty$, then $u = v = 0$ corresponds to a global minimum, otherwise, it is clear to see that there exists $g$ such that $1 - \|\xi_g\|^2 < 0$ and $v = 0$ is a strict saddle point.

*Remark* 2. Since $f \in C^\infty$, it is Lipschitz smooth on any bounded domain. As mentioned, $f$ is coercive when $R$ is coercive, and hence, its sublevel sets are bounded. So, for any descent algorithm, we can apply results based on $\nabla^k f$ being Lipschitz smooth for all $k$.

*Remark* 3. The nondegeneracy condition that $\|\xi_g\|_2 < 1$ outside the support of $v$ and invertibility of $(X_g^\top X_h)_{g,h \in J}$ is often used to derive consistency results [Bach, 2008, Zhao and Yu, 2006]. By Proposition 1, we see that this condition guarantees that the Hessian of $f$ is positive definite at the minimum, and hence, combining with the smoothness properties of $f$ explained in the previous remark, BFGS is guaranteed to converge superlinearly for starting points sufficiently close to the optimum [Nocedal and Wright, 2006, Theorem 6.6].

## 4 Numerical experiments

In this section, we use L-BFGS [Byrd et al., 1995] to optimise our bilevel function $f$ and we denote the resulting algorithm "Noncvx-Pro". Throughout, the inner problem is solved exactly using either a full or a sparse Cholesky solver. One observation from our numerics below is that although Noncvx-Pro is not always the best performing, unlike other solvers, it is robust to a wide range of settings: for example, our solver is mostly unaffected by the choice of $\lambda$ while one can observe in Figures 1 and 2 that this has a large impact on the proximal based methods and coordinate descent. Moreover, Noncvx-Pro is simple to code and rely on existing robust numerical routines (Cholesky/ conjugate gradient + BFGS) which naturally handle sparse/implicit operators, and we thus inherit their nice convergence properties. All numerics are conducted on 2.4 GHz Quad-Core Intel Core i5 processor with 16GB RAM. The code to reproduce the results of this article is available online[3].

---

[3] `https://github.com/gpeyre/2021-NonCvxPro`

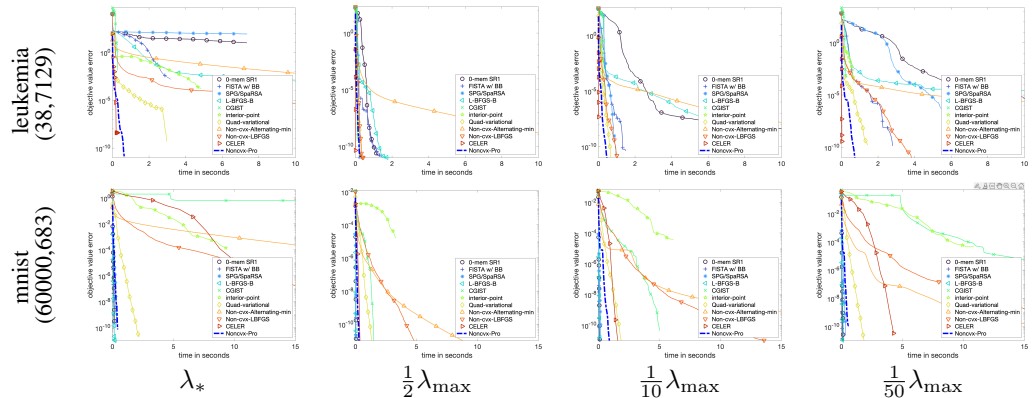

Figure 1: Lasso comparisons with different regularisation parameters.

## 4.1 Lasso

We first consider the Lasso problem where $R(\beta) = \sum_{i=1}^n |\beta_i|$.

**Datasets.** We tested on 8 datasets from the Libsvm repository[4]. These datasets are mean subtracted and normalised by $m$.

**Solvers.** We compare against the following 10 methods:

1. *0-mem SR1:* a proximal quasi newton method [Becker et al., 2019].
2. *FISTA w/ BB:* FISTA with Barzilai–Borwein stepsize [Barzilai and Borwein, 1988] and restarts [O'donoghue and Candes, 2015].
3. *SPG/SpaRSA:* spectral projected gradient [Wright et al., 2009].
4. *CGIST:* an active set method with conjugate gradient iterative shrinkage/thresholding [Goldstein and Setzer, 2010].
5. *Interior point method:* from Koh et al. [2007].
6. *CELER:* a coordinate descent method with support pruning [Massias et al., 2018].
7. *Non-cvx-Alternating-min:* alternating minimisation of $u$ and $v$ in $G$ from (3) [Hoff, 2017].
8. *Non-cvx-LBFGS:* Apply L-BFGS to minimise the function $(u, v) \mapsto G(u, v)$ in (3).
9. *L-BFGS-B* [Byrd et al., 1995]: apply L-BFGS-B under positivity constraints to $\min_{u,v \in \mathbb{R}_+^n} \sum_i u_i + \sum_i v_i + \frac{1}{2\lambda} \|X(u-v) - y\|_2^2$. This is the standard approach for applying L-BFGS to $\ell_1$ minimisation and corresponds to splitting $\beta$ into its positive and negative parts.
10. *Quad-variational:* Based on our idea of Noncvx-Pro, another natural (and to our knowledge novel) approach is to apply L-BFGS-B to the bilevel formulation of (1) without nonconvex reparametrization. Indeed, by applying (1) and using convex duality, the Lasso can solved by minimizing $g(\eta) \triangleq \max_{\alpha \in \mathbb{R}^m} \frac{1}{2} \sum_i \eta_i - \frac{\lambda}{2} \|\alpha\|^2 - \frac{1}{2} \sum_i \eta_i |\langle x_i, \alpha \rangle|^2 + \langle \alpha, y \rangle$. The gradient of $g$ is $g(\eta) = \frac{\lambda}{2} - \frac{1}{\lambda} |X^\top \alpha|^2$ where $|\cdot|^2$ is in a pointwise sense and $\alpha$ maximises the inner problem, and we apply L-BFGS-B with positivity constraints to minimise $g$.

**Experiments.** The results are shown in Figure 1 (with further experiments in the appendix). We show comparisons at different regularisation strengths, with $\lambda_*$ being the regularisation parameter found by 10 fold cross validation on the mean squared error, and $\lambda_{\max} = \|X^\top y\|_\infty$ is the smallest parameter at which the Lasso solution is guaranteed to be trivial.

## 4.2 Group Lasso

The multi-task Lasso [Gramfort et al., 2012] is the problem (7) where one minimises over $\beta \in \mathbb{R}^{n \times q}$, the observed data is $y \in \mathbb{R}^{m \times q}$ and $R(\beta) = \sum_{j=1}^n \|\beta^j\|_2$ with $\beta^j \in \mathbb{R}^q$ denotes the $j^{\text{th}}$ row of the matrix $\beta$ and the loss function is $\frac{1}{2} \|y - X\beta\|_F^2$ in terms of the the Frobenius norm.

---

4 https://www.csie.ntu.edu.tw/~cjlin/libsvmtools/datasets/

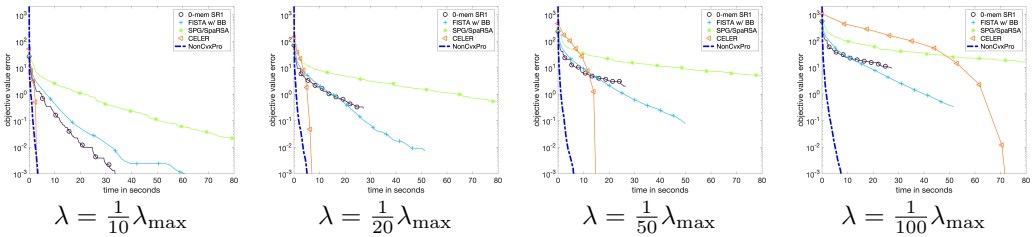

$$\lambda = \tfrac{1}{10}\lambda_{\max} \qquad \lambda = \tfrac{1}{20}\lambda_{\max} \qquad \lambda = \tfrac{1}{50}\lambda_{\max} \qquad \lambda = \tfrac{1}{100}\lambda_{\max}$$

Figure 2: Comparisons for multitask Lasso on MEG/EEG data

**Datasets.** We benchmark our proposed scheme on a joint MEG/EEG data from Ndiaye et al. [2015]. $X$ denotes the forward operator with $n = 22494$ source locations, and is constructed from 301 MEG sensors and 59 EEG sensors, so that $m = 360$. The observations $y \in \mathbb{R}^{m \times q}$ represent time series measurements at $m$ sensor with $q = 181$ timepoints each. The $\beta$ corresponds to the source locations which are assumed to remain constant across time.

**Solvers.** We perform a comparison against FISTA with restarts and BB step, the spectral projected gradient method SPG/SpaRSA, and CELER. Since $n$ is much larger than $q$ and $m$, we use for NonCvx-Pro the saddle point formulation (8) where the maximisation is over $\alpha \in \mathbb{R}^{m \times q}$. Computation of $\alpha$ in $\nabla f(v)$ involves solving $(\mathrm{Id}_m + \frac{1}{\lambda} X \operatorname{diag}(v^2) X^\top)\alpha = y.$, that is, $q$ linear systems each of size $m$.

**Experiments.** Figure 2 displays the objective convergence plots against running time for different $\lambda$: $\lambda = \lambda_{\max}/r$ with $\lambda_{\max} = \max_i \|X_i^\top y\|_2$ for $r = 10, 20, 50, 100$. We observe substantial performance gains over the benchmarked methods: In MEG/EEG problems, the design matrix tends to exhibit high correlation of columns and proximal-based algorithms tend to perform poorly here. Coordinate descent with pruning is known to perform well here when the regularisation parameter large [Massias et al., 2018], but its performance deteriorates as $\lambda$ increases.

### 4.3 Trace norm

In multi-task feature learning [Argyriou et al., 2008], for each task $t = 1, \ldots, T$, we aim to find $f_t : \mathbb{R}^n \to \mathbb{R}$ given training examples $(x_{t,i}, y_{t,i}) \in \mathbb{R}^n \times \mathbb{R}$ for $i = 1, \ldots, m_t$. One approach is to jointly solve these $T$ regression problems by minimising (5) where $R$ is the trace norm (also called "nuclear norm"), $r = T$, $y = (y_t)_{t=1}^T$, $\mathcal{A}(B) = (X_t B_t)_{t=1}^T$ with $X_t$ being the matrix with $i^{\text{th}}$ row as $x_{t,i}$ and $B_t$ being the $t^{\text{th}}$ column of the matrix $B$. Note that letting $u_t \in \mathbb{R}^n$ denote the $t^{\text{th}}$ column of $U$, the computation of $U$ in $\nabla f(V)$ involves solving $T$ linear systems $(\lambda \mathrm{Id}_n + V^\top X_t^\top X_t V)u_t = V^\top X_t^\top y_t$. Here, the trace norm encourages the tasks to share a small number of linear features.

**Datasets.** We consider the three datasets commonly considered in previous works. The *Schools dataset* [5] from Argyriou et al. [2008] consists of the scores of 15362 students from 139 schools. There are therefore $T = 139$ tasks with $15362 = \sum_t m_t$ data points in total, and the goal is to map $n = 27$ student attributes to exam performance. The *SARCOS dataset* [6] [Zhang and Yang, 2021, Zhang and Yeung, 2012] has 7 tasks, each corresponding to learning the dynamics of a SARCOS anthropomorphic robot arms. There are $n = 21$ features and $m = 48,933$ data points, which are shared across all tasks. The *Parkinsons dataset* [Tsanas et al., 2009] [7] which is made up of $m = 5875$ datapoints from $T = 42$ patients. The goal is to map $n = 19$ biomarkers to Parkinson's disease symptom scores for each patient.

**Solvers.** Figure 3 reports a comparison against FISTA with restarts and IRLS. The IRLS algorithm for (5) is introduced in Argyriou et al. [2008] (see also Bach et al. [2011]), and applies alternate minimisation after adding the regularisation term $\varepsilon \lambda \operatorname{tr}(Z^{-1})/2$ to (4). The update of $B$ is a simple least squares problem while the update for $Z$ is $Z \leftarrow (BB^\top + \varepsilon \mathrm{Id})^{\frac{1}{2}}$. Our nonconvex approach has

---

[5] https://home.ttic.edu/~argyriou/code/     [6] http://www.gaussianprocess.org/gpml/data/
[7] http://archive.ics.uci.edu/ml/datasets/Parkinsons+Telemonitoring

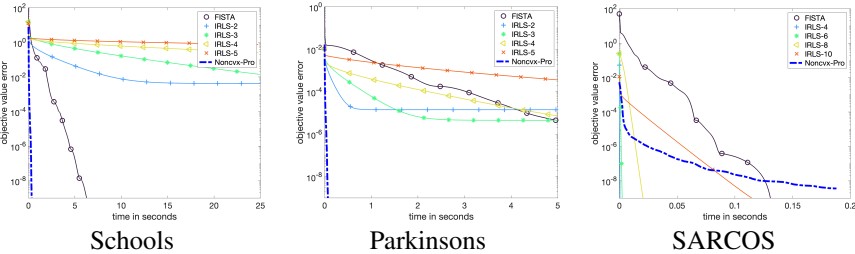

Schools         Parkinsons         SARCOS

Figure 3: Comparisons for multi-task feature learning, IRLS-d corresponds to IRLS with $\varepsilon = 10^{-d}$.

the same per-iteration complexity as IRLS, but one advantage is that we directly deal with (5) without any $\varepsilon$ regularisation.

*Remark* 4. Quad-variational mentioned in Section 4.1 does not extend to the trace norm case, since the function $g$ would be minimised over $\mathbb{S}_+^n$, for which the application of L-BFGS-B is unclear. For this reason, bilevel formulations for the trace norm [Pong et al., 2010] have been restricted to the use of first order methods.

**Experiments.**    For each dataset, we compute the regularisation parameter $\lambda_*$ by 10-fold cross-validation on the RMSE averaged across 10 random splits. Then, with this regularisation parameter, we compare Non-convex-pro, FISTA and IRLS with different choices of $\varepsilon$. The convergence plots are show in Figure 3. We observe substantial computational gains for the Schools and Parkinson's dataset. For the SARCOS dataset, IRLS performed the best, and even though Nonconvex-pro is comparatively less effective here, although we remark that the number of tasks is much smaller ($T = 7$) and the recorded times are much shorter (less than 0.2s). Further numerical illustrations with synthetic data are shown in the appendix – our method is typically less sensitive to problem variations.

## 4.4    Constraint Group Lasso and Optimal Transport

A salient feature of our method is that it can handle arbitrary small regularization parameter $\lambda$ and can even cope with the constrained formulation, when $\lambda = 0$, which cannot be tackled by most state of the art Lasso solvers. To illustrate this, we consider the computation of an Optimal Transport (OT) map, which has recently gained a lot of attention in ML [Peyré et al., 2019]. We focus on the Monge problem, where the ground cost is the geodesic distance on either a graph or a surface (which extends original Monge's problem where the cost is the Euclidean distance). This type of OT problems has been used for instance to analyze and process brain signals in M/EEG [Gramfort et al., 2015], for application in computer graphics [Solomon et al., 2014] and is now being applied to genomic datasets [Schiebinger et al., 2019]. As explained for instance in [Santambrogio, 2015, Sec.4.2], the optimal transport between two probability measures $a$ and $b$ on a surface can be computed by advecting the mass along a vector field $v(x) \in \mathbb{R}^3$ (tangent to the surface) with minimum vectorial $L^1$ norm $\int \|v(x)\| \mathrm{d}x$ (where $\mathrm{d}x$ is the surface area measure) subject to the conservation of mass $\mathrm{div}(v) = a - b$. Once discretized on a 3-D mesh, this boils down to solving a constrained group Lasso problem $(\mathcal{P}_0)$ where $\beta_g \in \mathbb{R}^3$ is a discretization of $v(z_g)$ at some vertex $z_g$ of the mesh, $X$ is a finite element discretization of the divergence using finite elements and $y = a - b$. The same applies on a graph, in which case the vector field is aligned with the edge of the graph and the divergence is the discrete divergence associated to the graph adjacency matrix, see Peyré et al. [2019]. This formulation is often called "Beckmann problem".

**Datasets.**    We consider two experiments: (i) following Gramfort et al. [2015] on a 3-D mesh of the brain with $n = 20000$ vertices with localized Gaussian-like distributions $a$ and $b$ (in blue and red), (ii) a 5-nearest neighbors graph in a 7-dimensional space of gene expression (corresponding to 7 different time steps) of baker's yeast, which is the dataset from DeRisi et al. [1997]. The $n = 614$ nodes correspond to the most active genes (maximum variance across time) and this results in a graph with 1927 edges. The distributions are synthetic data, where $a$ is a localized source whereas $b$ is more delocalized.

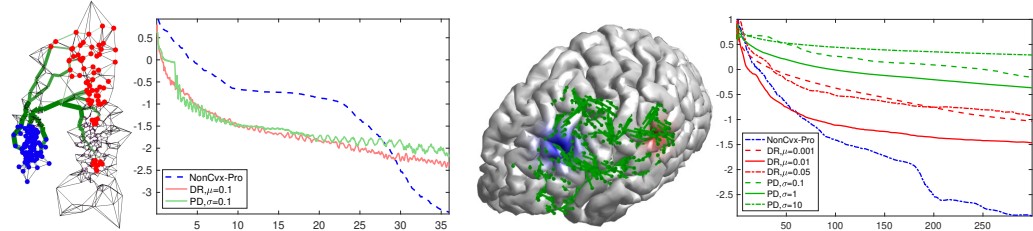

Figure 4: Resolution of Beckmann problem on a graph (left) and a 3-D mesh (right). The probability distributions $a$ and $b$ are displayed in blue and red and the optimal flow $\beta$ is represented as green segments (on the edge of the graph and on the faces of the mesh). The convergence curves display the decay of $\log_{10}(\|\beta_t - \beta^\star\|_1)$ during the iterations of the algorithms (DR=Douglas-Rachford, PD=Primal-Dual) as a function of time $t$ in second.

**Solvers.** We test our method against two popular first order schemes: the Douglas-Rachford (DR) algorithm [Lions and Mercier, 1979, Combettes and Pesquet, 2007] (DR) and the primal-dual scheme of Chambolle and Pock [2011]. DR is used in its dual formulation (the Alternating Direction Method of Multipliers – ADMM) but on a re-parameterized problem in Solomon et al. [2014]. The details of these algorithms are given in Appendix E.

**Experiments.** Figure 4 shows the solution of this Beckmann problem in these two settings. While DR has the same complexity per iteration as the computation of the gradient of $f$ (resolution of a linear system), a chief advantage of PD is that it only involves the application of $X$ and $X^\top$ et each iterations. Both DR and PD have stepping size parameters (denoted $\mu$ and $\sigma$) which have been tuned manually (the rightmost figure shows two other sub-optimal choices of parameters). In contrast, our algorithm has no parameter to tune and is faster than both DR and PD on these two problems.

## 5 Conclusion

Most existing approaches to sparse regularisation involve careful smoothing of the nonsmooth term, either by proximal operators or explicit regularisation as in IRLS. We propose a different direction: a simple reparameterization leads to a smooth optimisation problem, and allows for the use of standard numerical tools, such as BFGS. Our numerical results demonstrate that this approach is versatile, effective and can handle a wide range of problems.

We end by making some remarks on possible future research directions. The application of our method to other loss functions requires the use of an inexact solver for the inner problems, and controlling the impact of its approximation is an interesting avenue for future work. Furthermore, it is possible that one can obtain further acceleration by combining with screening rules or active set techniques.

## Acknowledgments

The work of G. Peyré was supported by the French government under management of Agence Nationale de la Recherche as part of the "Investissements d'avenir" program, reference ANR19-P3IA-0001 (PRAIRIE 3IA Insti- tute) and by the European Research Council (ERC project NORIA).

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
