# Supplementary to Smooth Bilevel Programming for Sparse Regularization

Clarice Poon,[*]    Gabriel Peyré[†]

## A    Pseudocode for gradient descent implementation

For concreteness, we write down in Algorithm 1 the gradient descent algorithm for solving

$$\min_{\beta \in \mathbb{R}^n} \frac{1}{2\lambda} \|X\beta - y\|_2^2 + \|\beta\|_1,$$

where we recall that $X \in \mathbb{R}^{m \times n}$. The choice of $\lambda = 0$ corresponds to the Basis-pursuit setting. Note that $\nabla f(\beta_t) = g_t$ is computed either as in line 5 or line 9 of the algorithm and one can use these computations for any gradient based algorithm (e.g. BFGS). Note also that this is simply gradient descent on a smooth function, and one can apply typical methods to choosing the stepsize $\gamma_k$, such as the Barzilai-Borwein stepsize [Barzilai and Borwein, 1988].

---

**Algorithm 1:** Gradient descent implementation of Ncvx-Pro for solving Lasso.

1  initialization $v_0 \in \mathbb{R}^n$ (with no zero entries), stepsize $\gamma_t > 0$;
   **Result:** $\beta_t$
2  **while** *not converged* **do**
3     **if** $n \leqslant m$ *and* $\lambda > 0$ **then**
4        $u_t = -\left(\mathrm{diag}(v_t)X^\top X \,\mathrm{diag}(v_t) + \lambda\mathrm{Id}\right)^{-1}\left(v_t \odot X^\top y\right);$
5        $g_t = v_t \odot v_t + \frac{1}{\lambda}u_t \odot X^\top(Xu_t \odot v_t - y);$
6        $\beta_t = u_t \odot v_t;$
7     **else**
8        $\alpha_t = -\left(X\,\mathrm{diag}(v_t \odot v_t)X^\top + \lambda\mathrm{Id}\right)^{-1}y;$
9        $g_t = v_t \odot v_t - v_t \odot |X^\top \alpha_t|^2;$
10       $\beta_t = -v_t \odot v_t \odot X^\top \alpha_t;$
11    **end**
12    $v_{t+1} = v_t - \gamma_t g_t$
13 **end**

---

## B    Proofs and additional results for Section 2

*Proof to Theorem 1.* To show that i) implies ii), recall that a convex, proper and lower semicontinuous function $-\varphi$ can be written in terms of its convex conjugate which has domain $\mathbb{R}^d_-$. By writing $\beta^2 \triangleq \beta \odot \beta$, using the definition of $R$, we have

$$-R(\beta^2) = -\varphi(\beta^2) = \sup_{v \leqslant 0}\langle \beta^2, v \rangle - (-\varphi)^*(v) = -\inf_{u \geqslant 0}\langle \beta^2, u \rangle + (-\varphi)^*(-u).$$

which is ii) with $\psi(u) \triangleq (-\varphi)^*(-u)$ as required.

---

[*]University of Bath, Bath BA2 7AY, UK cmshp20@bath.ac.uk    [†]CNRS and DMA, Ecole Normale Supérieure, 45 rue d'Ulm, F-75230 PARIS cedex 05, FRANCE, gabriel.peyre@ens.fr

35th Conference on Neural Information Processing Systems (NeurIPS 2021), Sydney, Australia.

Conversely, if $R$ is of the form in ii), then

$$R(\beta) = \inf_{u \in \mathbb{R}_+^n} \langle u, \beta^2 \rangle + \psi(u) = -\sup_{u \in \mathbb{R}_+^n} -\langle u, \beta^2 \rangle - \psi(u),$$

so $R(\beta) = -\psi^*(-\beta \odot \beta)$ and $-\psi^*(-\cdot)$ is clearly a proper, upper semicontinuous, concave function.

For the expression of $\psi$ when $R$ is a norm, from the above, we know that $\psi = (-\varphi)^*(-z)$, and recall that for any norm, $R(\beta) = \max_{R^*(w) \leqslant 1} \langle w, \beta \rangle$. So,

$$\psi(z) = \max_{u \geqslant 0} \langle -u, z \rangle + \varphi(u)$$

$$= \max_{\beta} \langle -\beta^2, z \rangle + \varphi(\beta^2) = \max_{\beta} \langle -\beta^2, z \rangle + R(\beta)$$

$$= \max_{\beta} \langle -z, \beta^2 \rangle + \max_{R^*(w) \leqslant 1} \langle \beta, w \rangle = \max_{R^*(w) \leqslant 1} \frac{1}{4} \sum_i \frac{w_i^2}{z_i},$$

where in the line, we swapped the two maximums and used the optimality condition over $\beta$ which is $2\beta \odot z = w$. That is, $h(\eta) = 2\psi(-\frac{1}{2\eta}) = \max_{R^*(w) \leqslant 1} \sum_i w_i^2 \eta_i$.

To derive the identity for $R(\beta)^2$, by the Cauchy-Schwarz inequality

$$R(\beta)^2 = \sup_{R^*(w) \leqslant 1} |\langle \beta, w \rangle|^2 \leqslant \sup_{R^*(w) \leqslant 1} \left( \sum_i \beta_i^2 \eta_i \right) \left( \sum_i \frac{w_i^2}{\eta_i} \right) = 4\psi(\eta) \sum_i \beta_i^2 \eta_i$$

for all $\eta > 0$. Therefore,

$$R(\beta)^2 \leqslant \inf_{\eta \geqslant 0, \psi(\eta) \leqslant \frac{1}{4}} \sum_i \beta_i^2 \eta_i$$

$$= \sup_{\lambda > 0} \inf_{\eta \geqslant 0} \lambda(\psi(\eta) - \frac{1}{4}) + \sum_i \beta_i^2 \eta_i$$

$$= \sup_{\lambda > 0} -\frac{\lambda}{4} + \lambda R(\beta/\sqrt{\lambda}) = \sup_{\lambda > 0} -\frac{\lambda}{4} + \sqrt{\lambda} R(\beta) = R(\beta)^2.$$

where we used the identity $\lambda R(\beta/\sqrt{\lambda}) = \sum_i \beta_i^2 \eta_i + \lambda \psi(\eta)$ and the fact that $R$ is one positive homogeneous.

$\square$

We derive some properties of the function $h$:

**Lemma 1.** *Consider the function $\varphi$ and $\psi$ from Theorem 1. If $\varphi : [0, \infty) \to [L, U]$, where $L > -\infty$ and $U \in \mathbb{R} \cup \{+\infty\}$, then $\psi$ is an decreasing function with domain contained in $[0, \infty)$, taking values in $[L, U]$. If $R$ is coercive, then $\lim_{\|z\| \to 0} \psi(z) = +\infty$.*

*Proof to Lemma 1.* Let $\varphi : [0, \infty) \to [L, U]$, where $L > -\infty$ and $U \in \mathbb{R} \cup \{+\infty\}$. We describe the properties of the function $\psi(z) = (-\varphi)^*(-z) = \sup_{u \geqslant 0} \langle z, -u \rangle + \varphi(u)$.

(i) $\operatorname{dom}(\psi) \subset [0, \infty)$: since $\varphi$ is bounded below, it is clear that for $z < 0$, $\sup_{u \geqslant 0} \langle z, -u \rangle + \varphi(u) = +\infty$.

(ii) $\psi(0) = \sup_{u \geqslant 0} \varphi(u) = U$.

(iii) Suppose $M \triangleq \sup \{v \setminus v \in \partial\varphi(u), u \geqslant 0\} < \infty$. Then, for all $z \geqslant M$, $-z + \partial\varphi(u) < 0$ for all $u \geqslant 0$ and hence, $\psi(z) = \varphi(0) \geqslant L$. Therefore, $\psi$ takes values in $[L, U]$. So, if $M$ is finite, then one can restrict the optimisation over $z$ to values in $[0, M]$.

(iv) $\psi$ is decreasing: By Danskin's theorem [Bertsekas, 1997, Prop. B.25], for $z \in \{v \setminus v \in \partial\varphi(u), u \geqslant 0\}$,

$$\partial\psi(z) = \{-u \setminus u \in \operatorname{argmin}_{u \geqslant 0} \langle u, -z \rangle + \varphi(u)\} \subset (-\infty, 0).$$

$\square$

## B.1 Functions on matrices

We have the following result for matrix valued functions. Let $\mathbb{S}^n_+$ denote the set of symmetric positive semidefinite matrices.

**Theorem 1.** *Let $R : \mathbb{R}^{n \times m} \to \mathbb{R}$. The following are equivalent*

    *i) $R(B) = \varphi(BB^\top)$ where $\varphi$ is a proper concave upper semi-continuous function with domain $\mathbb{S}^n_+$.*

    *ii) There exists a convex function $\psi$ such that $R(B) = \min_{Z \in \mathbb{S}^n_+} \operatorname{tr}(B^\top Z B) + \psi(Z)$.*

*Moreover, we have $\psi(Z) = (-\varphi)^*(-Z)$. If $R$ is a norm, then $\psi$ can be written as*

$$\psi(Z) = \max_{R^*(W) \leqslant 1} \frac{1}{4} \operatorname{tr}(W^\top Z^{-1} W). \tag{1}$$

*Moreover,*

$$R(B)^2 = \inf_{Z \in \mathbb{S}^n_+} \left\{ \operatorname{tr}(B^\top Z B) \setminus \psi(Z) \leqslant \frac{1}{4} \right\}. \tag{2}$$

**Nuclear norm** $R(W) = \operatorname{tr}(\sqrt{WW^\top})$ where $\sqrt{\cdot}$ is the matrix square root. On the space of symmetric positive semidefinite matrices, $\varphi(B) = \operatorname{tr}(\sqrt{B})$ is concave and $\psi(D) = \frac{1}{4}\operatorname{tr}(D^{-1})$, where we use $\partial_A \operatorname{tr}(\sqrt{A}) = (2\sqrt{A})^{-1}$ for all symmetric positive semidefinite matrices and $\partial_A \operatorname{tr}(AB) = B$.

**(Nonconvex) spectral regularisation** Given a symmetric psd matrix $Z = U \operatorname{diag}(\sigma_i) U^\top$ and $\alpha > 0$, let $Z^\alpha \triangleq U \operatorname{diag}(\sigma_i^\alpha) U^\top$. For $\alpha \in (0,1)$, consider $R(W) = \operatorname{tr}((WW^\top)^{\alpha/2}) = \sum_i \sigma_i^\alpha$ where $\sigma_i$ are the singular values of $W$. Then, given a symmetric psd matrix, $\varphi(Z) = \operatorname{tr}(Z^{\alpha/2})$ which is concave [Bhatia, 2009, Thm 4.2.3] and

$$\psi(Z) = \min_{V \in \mathbb{S}^d_+} -\operatorname{tr}(VZ) + \varphi(V) = \min_{U \in \mathbb{O}^d, \sigma \in \mathbb{R}^d_+} -\operatorname{tr}(\operatorname{diag}(\sigma) U Z U^\top) + \sum_i \sigma_i^{\alpha/2}$$

$$= \min_{U \in \mathbb{O}^d, \sigma \geqslant 0} -\sum_i \hat{Z}_{ii} \sigma_i + \sum_i \sigma_i^{\alpha/2} \quad \text{where} \quad \hat{Z} = U Z U^\top$$

$$= C_\alpha \min_{U \in \mathbb{O}^d} \sum_i \hat{Z}_{ii}^{\frac{\alpha}{\alpha-2}} \quad \text{where} \quad \hat{Z} = U Z U^\top \quad \text{and} \quad U \in \mathbb{O}^d$$

$$= C_\alpha \operatorname{tr}(Z^{\frac{\alpha}{\alpha-2}})$$

Therefore,

$$R(B) = \inf_{Z \in \mathbb{S}^d_+} \operatorname{tr}(B^\top Z B) + C_\alpha \operatorname{tr}(Z^{\frac{\alpha}{\alpha-2}}).$$

*Proof of Theorem 1.* To derive (1),

$$\psi(Z) = \max_{U \in \mathbb{S}^n_+} -\langle U, Z \rangle + \varphi(U) = \max_{V \in \mathbb{R}^n} -\langle VV^\top, Z \rangle + \varphi(VV^\top)$$

$$= \max_{V \in \mathbb{R}^n} -\langle VV^\top, Z \rangle + R(V)$$

Then, (1) follows, since by convex duality and definition of $R^*$,

$$\max_{R^*(W) \leqslant 1} \frac{1}{4} \operatorname{tr}(W^\top Z^{-1} W) = \max_{R^*(W) \leqslant 1} \max_V \langle -Z, VV^\top \rangle + \langle V, W \rangle = \max_V \langle -Z, VV^\top \rangle + R(V).$$

Finally, by the submultiplicative property of the Frobenius norms, for all $Z \in \mathbb{S}^n_+$ with $Z \succ 0$,

$$R(B)^2 = \sup_{R^*(W) \leqslant 1} |\langle Z^{-1/2} W, Z^{1/2} B \rangle|^2 \leqslant \sup_{R^*(W) \leqslant 1} \operatorname{tr}(W^\top Z^{-1} W) \operatorname{tr}(B^\top Z B)$$

$$= 4\psi(W) \operatorname{tr}(B^\top Z B)$$

It follows that just as in the proof of Theorem 1 that

$$R(B)^2 \leqslant \inf_{Z \in \mathbb{S}^n_+} \operatorname{tr}(B^\top Z B) \quad \text{where} \quad \psi(Z) \leqslant \frac{1}{4}.$$

$\square$

## C    Proof of Section 3

*Proof of Proposition 3.*  Let

$$G(u,v) \triangleq \frac{1}{2}\|u\|^2 + \frac{1}{2}\|v\|_2^2 + \frac{1}{2\lambda}\|X(v \odot_{\mathcal{G}} u) - y\|_2^2.$$

We know from Theorem 2 that $f$ is differentiable with

$$\nabla f(v) = \partial_v G(u,v) = v + \lambda^{-1} u \odot X^\top (Xv \odot_{\mathcal{G}} u - y)$$

where $u = \mathrm{argmin}_u \, G(u,v)$. In particular,

$$0 = \partial_u G(u,v) = u + \lambda^{-1} X^\top (X(v \odot_{\mathcal{G}} u) - y).$$

Since $\partial_{uu} G = \lambda^{-1}(v_g X_g^\top X_h v_h)_{g,h} + \mathrm{Id}$ is invertible, by the implicit function theorem $u$ is a smooth function of $v$ with $\partial_v u = [\partial_{uu} G]^{-1} \partial_{vu} G$. In particular,

$$\nabla^2 f(v) = \partial_{vv} G(u,v) + \partial_{uv} G(u,v) \partial_v u.$$

So, the Hessian of $f$ is the Schur complement of the Hessian of $G$ (as also observed in Ruhe and Wedin [1980], van Leeuwen and Aravkin [2016]). We write $\nabla^2 G = \begin{pmatrix} A & B \\ B^\top & D \end{pmatrix}$ where

$$A \triangleq \partial_{vv} G = \lambda^{-1} \left( u_g^\top X_g^\top X_h u_h \right)_{g,h} + \mathrm{Id}$$

$$B \triangleq \partial_{uv} G = \lambda^{-1} \left( (u_g^\top X_g^\top X_h v_h)_{g,h} \right) + \mathrm{diag}(\xi_g^\top)$$

$$D \triangleq \partial_{uu} G = \lambda^{-1}(v_g X_g^\top X_h v_h)_{g,h} + \mathrm{Id}$$

where $\xi = \frac{1}{\lambda} X^\top (X(u \odot v) - y)$. Then, $\nabla^2 f(t) = A - BD^{-1}B^\top$. Note that in fact, $u$ is infinitely differentiable by the implicit function theorem, and so, $f$ is also infinitely differentiable.

We now derive a formula for the Hessian of $f$. By permuting the rows and columns of $\nabla^2 G$, we can assume that, letting $J$ denote the support of $v$,

$$A \triangleq \begin{pmatrix} \lambda^{-1} \left( u_g^\top X_g^\top X_h u_h \right)_{g,h \in J} + \mathrm{Id}_J & 0 \\ 0 & \mathrm{Id}_{J^c} \end{pmatrix}$$

$$B \triangleq \lambda^{-1} \begin{pmatrix} ((u_g^\top X_g^\top X_h v_h)_{g,h \in J} + \lambda \, \mathrm{diag}(\xi_g^\top)_{g \in J}) & 0 \\ 0 & \lambda \, \mathrm{diag}(\xi_g^\top)_{g \in J^c} \end{pmatrix}$$

$$D \triangleq \begin{pmatrix} \lambda^{-1}(v_g X_g^\top X_h v_h)_{g,h \in J} + \mathrm{Id}_J & 0 \\ 0 & \mathrm{Id}_{J^c} \end{pmatrix}$$

Note that $A$ and $D$ is positive definite. So, $\nabla^2 G$ is positive semidefinite if and only if $A - BD^{-1}B^\top$ is positive semidefinite. Note that $A - BD^{-1}B^\top$ is a block diagonal matrix, with the bottom right block as $\mathrm{Id}_{J^c} - \mathrm{diag}(\|\xi_g\|_2^2)_{g \in J^c}$.

To work out the expression for the top left block of $A - BD^{-1}B^\top$, let us first examine the top left block of the matrix $B$: Note that by definition of $u$,

$$\lambda^{-1} v_g X_g^\top (X(v \odot_{\mathcal{G}} u) - y) + u_g = 0 \implies \forall g \in \mathrm{Supp}(v), \ \xi_g = -\frac{u_g}{v_g}.$$

Define the block diagonal matrix $U_{u/v} = \mathrm{diag}(u_g/v_g)_{g \in J}$, then $U_{u/v}^\top U_{u,v} = \mathrm{diag}(\|u_g\|^2/v_g^2)_{g \in J}$. The top left block of $B$ is

$$\lambda^{-1}(u_g^\top X_g^\top X_h v_h)_{g,h \in J} + \mathrm{diag}(\xi_g^\top)_{g \in J} = \lambda^{-1} U_{u/v}^\top (v_g X_g^\top X_h v_h)_{g,h \in J} + \mathrm{diag}(\xi_g^\top)$$

$$= U_{u/v}^\top \left( \lambda^{-1}(v_g X_g^\top X_h v_h)_{g,h \in J} + \mathrm{Id}_J \right) - U_{u/v}^\top + \mathrm{diag}(\xi_g^\top)$$

$$= U_{u/v}^\top \underbrace{\left( \lambda^{-1}(v_g X_g^\top X_h v_h)_{g,h \in J} + \mathrm{Id}_J \right)}_{\triangleq H} - 2 U_{u/v}^\top = U_{u/v}^\top W - 2 U_{u/v}^\top.$$

Therefore, the top left block of $D^{-1}$ is $H^{-1}$. So, the top left block of $BD^{-1}B^\top$ is

$$(U_{u/v}^\top H - 2U_{u/v}^\top)(U_{u/v} - 2H^{-1}U_{u/v}) = U_{u/v}^\top H U_{u/v} - 4U_{u/v}^\top U_{u/v} + 4U_{u/v}^\top H^{-1} U_{u/v}.$$

and the top left block of $A - BD^{-1}B^\top$ is

$$\mathrm{Id}_J - U_{u/v}^\top U + 4U_{u/v}^\top U_{u/v} - 4U_{u/v}^\top H^{-1}U_{u/v}$$

$$= \mathrm{diag}(1 - \|\xi_g\|_2^2)_{g \in J} + 4U_{u/v}^\top U_{u/v} - 4U_{u/v}^\top H^{-1}U_{u/v}.$$

Note that $\|\mathrm{Id} - H^{-1}\| \leqslant 1$, and given $w \in \mathbb{R}^{|\mathcal{G}|}$,

$$\langle \nabla^2 f(v)w,\, w \rangle = \sum_{g \in \mathcal{G}}(1 - \|\xi_g\|_2^2)w_g^2 + 4\langle(\mathrm{Id} - H^{-1})(w \odot_{\mathcal{G}} \xi),\, w \odot_{\mathcal{G}} \xi \rangle$$

$$\leqslant \sum_{g \in \mathcal{G}}(1 - \|\xi_g\|_2^2)w_g^2 + 4\|\xi_g\|_2^2 w_g^2,$$

and it follows that $\|\nabla^2 f(v)\| \leqslant 1 + 3\max_{g \in \mathcal{G}}\|\xi_g\|_2^2$. We have a global Lipschitz bound on the gradient of $f$ if $\|\xi_g\| \leqslant L$ for some $L$, which is true because for each $v$, $u$ minimises

$$\min_u \frac{1}{2}\|u\|_2^2 + \frac{1}{2\lambda}\|X(v \odot_{\mathcal{G}} u) - y\|_2^2 \leqslant \frac{\|y\|_2^2}{2\lambda}$$

So, $\max_{g \in \mathcal{G}}\|\xi_g\|_2 \leqslant \|y\|_2 \max_{g \in \mathcal{G}}\|X_g\|/\lambda$, and $\|\nabla^2 f(v)\| \leqslant 1 + 3\|y\|^2 \max_{g \in \mathcal{G}}\|X_g\|^2/\lambda^2$.

At stationary points, we also have

$$u_g^\top X_g^\top (X(v \odot_{\mathcal{G}} u) - y) + \lambda v_g = 0 \implies \forall g \in \mathrm{Supp}(v),\ u_g^\top \xi_g = -v_g.$$

Together, this means that at stationary points, $\|u_g\|^2 = v_g^2$ and $U_{u/v}^\top U_{u/v} = \mathrm{Id}_J$. Therefore, the top left block of $A - BD^{-1}B^\top$ becomes

$$4\mathrm{Id}_J - 4U_{u/v}^\top\left(\lambda^{-1}(v_g X_g^\top X_h v_h)_{g,h \in J} + \mathrm{Id}_J\right)^{-1}U_{u/v} \succeq 0$$

since $\lambda^{-1}(v_g X_g^\top X_h v_h)_{g,h \in J} + \mathrm{Id}_J \succeq (1 + \mu)\mathrm{Id}$, where $\mu = \min\mathrm{Eig}\left(\lambda^{-1}(v_g X_g^\top X_h v_h)_{g,h \in J}\right)$. Therefore, the smallest eigenvalue of $A - BD^{-1}B$ is at least

$$\lambda\min\left(4\mu/(1 + \mu), \min_{g \notin J}(1 - \|\xi_g\|^2)\right) \geqslant \min_{g \notin J}(1 - \|\xi_g\|^2)$$

Moreover, if $A - BD^{-1}B \succeq 0$, then $\min_{g \notin J}(1 - \|\xi_g\|^2) \geqslant 0$, which implies that $(u, v)$ defines a minimiser to the original group Lasso problem, hence, $(u, v)$ defines a global minimum. Therefore, every stationary point is either a global minimum or a strict saddle point.

$\square$

**Remarks on the comparison with ISTA in the introduction** To explain the observed behavior, note that gradient descent for $f$ with stepsize $\gamma$ reads $v_{k+1} = v_k - \gamma\nabla f(v_k) = v_k(1 - \gamma(1 - |\xi_k|^2))$ where $\xi_k \triangleq \frac{1}{\lambda}X^\top(Xv_k \odot u_k - y)$ (see Proposition 3). Note that if $\beta_*$ is a minimiser, then $\xi_* \triangleq \frac{1}{\lambda}X^\top(X\beta_* - y)$ satisfies $\|\xi_*\|_\infty \leqslant 1$ and the set $\{i \setminus |(\xi_*)_i| = 1\}$ is often called the extended support and contains the support of $\beta_*$. It is clear that we can expect coefficients outside the extended support to (eventually) decay to 0 geometrically. Since $\xi_k$ is uniformly bounded (see Proposition 3), for $\gamma$ sufficiently small, $v_k$ never changes sign and any sign change in the iterate $\beta_k \triangleq v_k \odot u_k$ is due to $u_k$. In contrast, the ISTA dynamics is $\beta_{k+1} = \mathrm{sign}(\beta_k - \gamma\xi_k)\max(|\beta_k - \gamma\xi_k| - \gamma, 0)$. Due to the thresholding operation, a coefficient of $\beta_k$ is initialised with the wrong sign will spend some iterations as 0 before correcting its sign.

## D  Supplementary to Section 4

### D.1  Remarks on numerical experiments

**Initialisation points** We generated random initialisation point from the normal distribution. In our experiments, methods which are not reparameterized (e.g. the proximal methods), are given the same random initial point, while reparameterized methods have their own random initialisation, since some of these require positive starting points and some need double the number of variables. We find that the comparisons are not much affected by the choice of initial points.

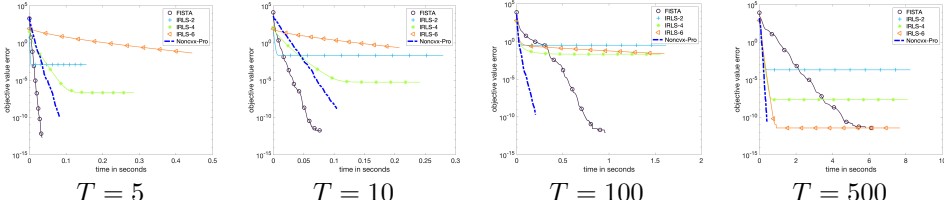

|  |  |  |  |
|---|---|---|---|
| $T = 5$ | $T = 10$ | $T = 100$ | $T = 500$ |

Figure 1: Multitask feature learning (nuclear norm regularisation) with synthetic data. We have $T$ tasks, $n = 30$ features and $m = 10,000$ samples in total. The matrix $X_t$ associated to each task has iid entries drawn uniformly at random from $[0, 1]$. See the description in Section 4.3.

**Inversion of linear systems**   As mentioned in Corollary (1), for the Lasso, when computing the gradient of $f$, one can either invert a $n \times n$ linear system or an $m \times m$ linear system. The same applies to Quad-variational, since the solution to the inner maximisation problem is, by the Woodbury identity,

$$\alpha = (\lambda \mathrm{Id}_m + X_\eta X_\eta^\top)^{-1} y = \frac{1}{\lambda} y - \frac{1}{\lambda} X_\eta (\lambda \mathrm{Id}_n + X_\eta^\top X_\eta)^{-1} (X_\eta^\top y)$$

where $X_\eta = X \operatorname{diag}(\sqrt{\eta})$, with the correspondence that $\beta = \eta \odot X^\top \alpha$. Throughout, we simply use backslash in MATLAB for the matrix inversion.

**Implementation details**   All numerics are done in Matlab with the exception of CELER which is in Python:

- CELER are conducted in Python and we used the code `https://mathurinm.github.io/celer/` provided by the original paper Massias et al. [2018]

- 0-mem SR1, FISTA w/ BB and SPG/SpaRSA use the Matlab code from `https://github.com/stephenbeckr/zeroSR1` of the paper Becker et al. [2019].

- Interior point method uses the Matlab code `https://web.stanford.edu/~boyd/l1_ls/` of Koh et al. [2007].

- CGIST uses the Matlab code `http://tag7.web.rice.edu/CGIST.html` of Goldstein and Setzer [2010].

- We had our own implementation of Non-cvx-Alternating-min and IRLS.

- Quad-variational, Non-cvx-LBFGS and Noncvx-Pro are written in Matlab using the L-BFGS-B solver from `https://github.com/stephenbeckr/L-BFGS-B-C` which is a Matlab wrapper for C code converted from the well known Fortran implementation of Byrd et al. [1995].

## D.2   Additional examples

**Lasso**   In Figure 2, we show additional numerics for the Lasso, testing against datasets from the Libsvm repository. The regularisation parameter $\lambda$ associated to each plots is found by cross validation on the mean squared error.

**Group Lasso**   In Figure 3, we show additional numerics for the multitask Lasso setup described in Section 4.2. We test on two synthetic datasets of size $(m, n, q) = (300, 1000, 100)$ with 5 relevant features and $(m, n, q) = (50, 1200, 20)$ with 10 relevant features. The data matrix $X$ has entries drawn from a normal distribution. We also test on a MEG/EEG dataset with $(m, n, q) = (305, 22494, 85)$ from the MNE repository `https://mne.tools/0.11/manual/datasets_index.html`. We display convergence plots for different regularisation parameters.

**Trace norm**   In Figure 1 we show additional numerics for the multifeature learning setup described in Section 4.3. The data matrices $X_t$ has entries drawn uniformly at random from $[0, 1]$. We consider different number of tasks $T$ tasks, $n = 30$ features and $m = 10,000$ samples in total (the samples are split at random across the different tasks).

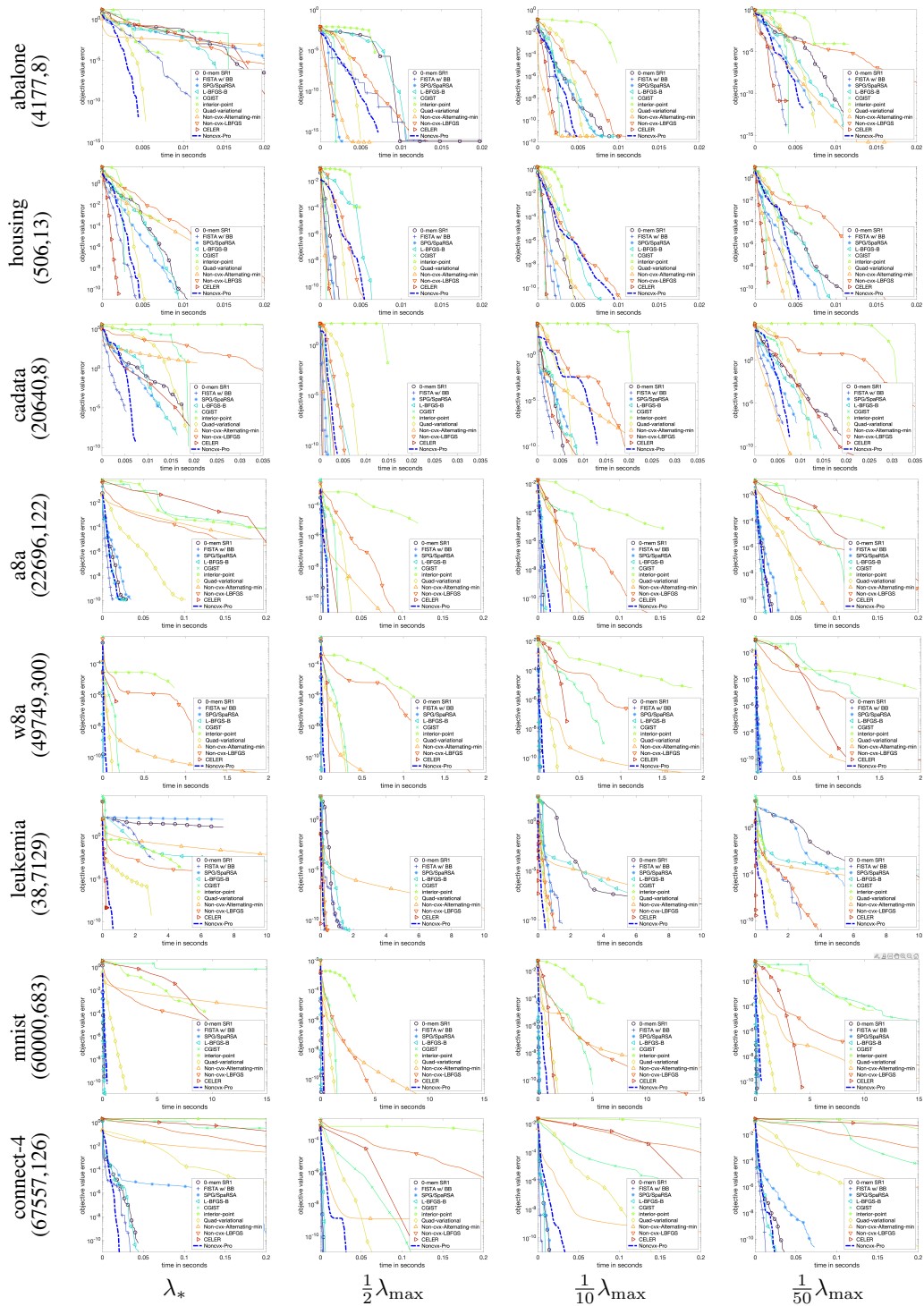

Figure 2: Comparisons of Lasso with different regularisation parameters on datasets from Libsvm. The first column shows the optimal regularisation parameter $\lambda_*$ found by cross validation. The second, third and fourth columns correspond to different fractions of $\lambda_{\max} = \|X^\top y\|_\infty$ which is the parameter for which the Lasso solution is identically zero. The smaller this fraction, the less sparse the solution.

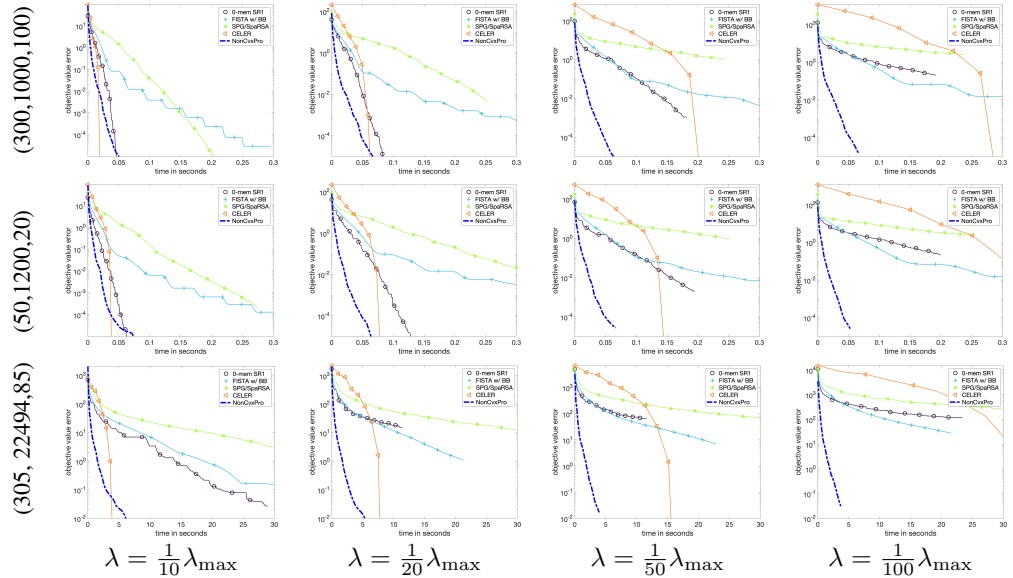

Figure 3: Comparisons for multitask Lasso at different regularisation strengths. The problem sizes $(m, n, q)$ are displayed on the left of each row. The top two rows are synthetic datasets generated by random Gaussian variables with 5 and 10 active features respectively. The last row corresponds to a MEG/EEG dataset from the MNE website

# E    Douglas-Rachford and Primal-Dual Algorithms

We consider the resolution of a constrained group Lasso problem

$$\min_{X\beta=y} \|\beta\|_{1,2} = \sum_g \|\beta_g\|_2$$

which we write as the minimization of either $F(\beta) + G(\beta)$ (for DR) or $F(\beta) + G_0(X\beta)$ where $F = \|\cdot\|_{1,2}$, $G = \iota_{\mathcal{C}}$ where the constraint set is $\mathcal{C} = \{\beta \setminus X\beta = y\}$ and $G_0 = \iota_{\{y\}}$. Here $\iota_{\mathcal{C}}$ is the convex indicator function of a closed convex set $\mathcal{C}$.

DR and PD are generic algorithm to solve minimization of function of the form $F+G$ and $F+G_0\circ X$ when one is able to compute efficiently the so-called proximal operator of the involved functionals, where the proximal operator of some convex function $H$ and some step size $\tau \geqslant 0$ is

$$\text{Prox}_{\tau H}(\beta) \triangleq \underset{\beta'}{\operatorname{argmin}} \frac{1}{2}\|\beta - \beta'\|_2^2 + H(\beta').$$

In our special case, one has

$$\text{Prox}_{\tau F}(\beta) = \left( \max(\|\beta_g\| - \tau, 0)\frac{\beta_g}{\|\beta_g\|} \right)_g, \quad \text{Prox}_{\tau G}(\beta) = \beta + X^\top(XX^\top)^{-1}(y - X\beta),$$

and $\text{Prox}_{\tau G_0}(\beta) = y$.

**DR algorithm.**    We denoted the reflected proximal map as $\text{rProx}_{\tau H}(\beta) = 2\,\text{Prox}_{\tau H}(\beta) - \beta$. For some step size $\mu > 0$ and weight $0 < \gamma < 2$ (which is set to $\gamma = 1$ in our experiments), the iterates $(\beta_k)_k$ of DR are $\beta_k \triangleq \text{Prox}_{\mu G}(z_k)$ where $z_k$ satisfies

$$z_{k+1} = (1 - \frac{\gamma}{2})z_k + \frac{\gamma}{2}\,\text{rProx}_{\mu F}(\text{rProx}_{\mu G}(z_k)).$$

**PD algorithm.**    Denoting $G_0^*(u) = \sup_\beta \langle \beta, u \rangle - G_0(\beta)$ the Legendre transform of $G_0$, the PD iterations read

$$w_{k+1} = \text{Prox}_{\sigma G_0^*}(w_k + \sigma X(\tilde{\beta}_k))$$
$$\beta_{k+1} = \text{Prox}_{\tau F}(\beta_k - \tau K^\top(w_{k+1}))$$
$$\tilde{\beta}_{k+1} = \beta_{k+1} + \theta(\beta_{k+1} - \beta_k).$$

In our case, one has $G_0^*(u) = \langle u, \, y \rangle$ so that $\mathrm{Prox}_{\sigma G_0^*}(u) = u - \tau y$. Convergence of the PD algorithm is ensure as long as $\tau \sigma \|X\|^2 < 1$ where $\|X\|$ is the operator norm, and $0 < \theta \leqslant 1$ (we use $\theta = 1$ in the numerical simulation). In our numerical simulation, we set $\tau \sigma \|X\|^2 = 0.9$ and tuned the value of the parameter $\sigma$.

## F  Non-convex optimisation with $\ell_q$ quasi-norms

As mentioned, for $q \in (0, 2)$, $R(\beta) \triangleq \|\beta\|_q^q = \sum_j |\beta_j|^q$ has a quadratic variational form. In the case where $q > 2/3$, we have the following bilevel smooth formulation:

**Corollary 1.** *When $q > 2/3$, $(\mathcal{P}_\lambda)$ is equivalent to*

$$\inf_{v \in \mathbb{R}^n} f(v) \triangleq \inf_{u \in \mathbb{R}^n} \frac{1}{2} \|u\|_2^2 + \frac{C_q}{2} \sum_{j=1}^n |v_j|^{\frac{2q}{2-q}} + \frac{1}{\lambda} L(X(u \odot v), y) \tag{3}$$

*where $C_q = (2-q)q^{q/(2-q)}$. The function $f$ is differentiable function provided that $q > 2/3$. Its gradient can be computed as in Theorem 2.*

*Remark* 1 (Existing approaches). Existing approaches to $\ell_q$ minimisation are typically iterative thresholding/proximal algorithms Bredies et al. [2015], IRLS Chartrand and Staneva [2008], Daubechies et al. [2004] or iterative reweighted $\ell_1$ algorithms Foucart and Lai [2009]. Iterative thresholding algorithms are applicable only for the case where the loss function is differentiable, and hence not applicable for Basis pursuit problems which we describe below. Moreover, computation of the proximal operation requires solving a nonlinear equation. For iterative reweighted algorithms, they require gradually decreasing an additional regularisation parameter $\varepsilon > 0$. This can be problematic in practice and for finite $\varepsilon$, one does not solve the original optimisation problem.

*Remark* 2. Since we have a differentiable unconstrained problem, the problem (3) can be handled using descent algorithms and convergence analysis is standard. For example, since $f$ is coercive, for any descent algorithm applied to $f$, we can assume that the generated sequence $v_k$ is uniformly bounded and $\nabla f(v_k)$ is also uniformly bounded. So, by applying standard results [Bertsekas, 1997, Proposition 1.2.1], we can conclude that all limit points of sequences $v_k$ generated by descent methods under line search on the stepsize are stationary points. In fact, since we have an unconstrained minimisation problem with a continuously differentiable $f$ which is also semialgebraic (for rational $q$) and hence satisfy the KL inequality Attouch et al. [2013], convergence of the full sequence by descent methods with line search can be guaranteed Noll and Rondepierre [2013].

### F.1  Basis pursuit

In this section, we focus on the basis pursuit problem with $q \in (2/3, 1)$,

$$\min_\beta \|\beta\|_q^q \quad \text{where} \quad X\beta = y.$$

The set of local minimums are all $\beta$ for which $X\beta = y$ and there exists $\alpha$ such that $\left(X^\top \alpha\right)_i = q|\beta_i|^{q-1}\mathrm{sign}(\beta_i)$ on the support of $\beta$. When $q > 2/3$ and $f$ is differentiable with

$$\nabla f(v) = q^{\frac{2}{2-q}} |v|^{\gamma-1}\mathrm{sign}(v) - v \odot |X^\top \alpha|^2, \quad \text{where} \quad \gamma \triangleq 2q/(2-q) > 1.$$

At a stationary point $v$, letting $\beta = -v^2 \odot X^\top \alpha$, we have $X\beta = y$ and $\nabla f(v) = 0$ implies that on the support of $v$, $q|v|^2 = |\beta|^{2-q}$ and so,

$$X^\top \alpha = -v^{-2}\beta = -q\,\mathrm{sign}(\beta) \odot |\beta|^{q-1},$$

which is precisely the optimality condition of the original problem.

**Illustrations for Basis pursuit**   In Figure 4, we show that gradient descent dynamics for $f$ in the case of the indicator function $L(\cdot, y) = \iota_{\{y\}}$ and a random Gaussian matrix $X \in \mathbb{R}^{10 \times 20}$, that is

$$v_{k+1} = v_k - \tau \nabla f(v_k) = v_k - \tau \left( q^{\frac{2}{2-q}} |v_k|^{\frac{3q-2}{2-q}} \odot \mathrm{sign}(v) - v_k \odot |X^\top \alpha_k|^2 \right)$$

where

$$X \,\mathrm{diag}(v_k \odot v_k)X^\top \alpha_k = -y.$$

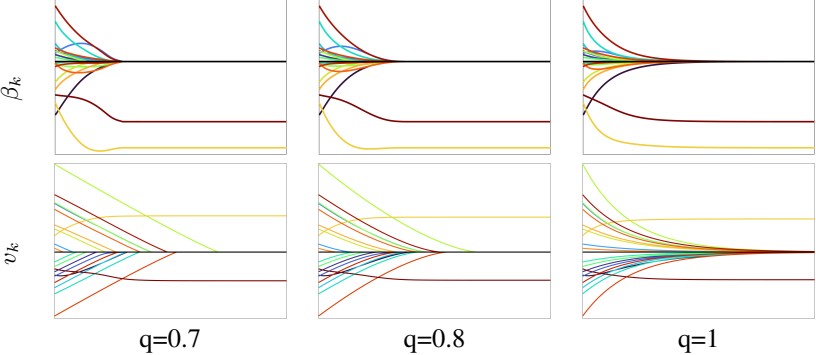

q=0.7        q=0.8        q=1

Figure 4: Evolution of 20 coefficients for Basis pursuit with $\ell_q$ regularisation. The same stepsize $\tau$ is used for all plots. Top row show the evolution of $\beta_k$ and the bottom row show the evolution of $v_k$.

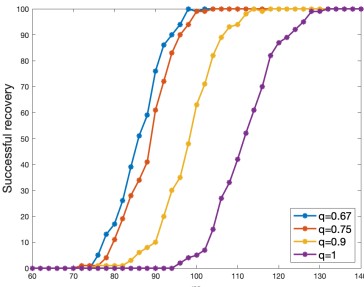

Figure 5: Number of successful recovery by $\ell_q$ minimisation.

Observe that as $q \to 2/3$, the evolution paths of $v_k$ becomes increasingly linear.

In Figure 5, we follow the experiment setup of Chartrand and Staneva [2008] and generate 100 problem instances $(\bar{X}, \bar{y}, \bar{\beta})$. Each problem instance consist of a matrix $\bar{X} \in \mathbb{R}^{m \times n}$ with $m = 140$ rows and $n = 256$ columns whose entries are identical independent distributed Gaussian random variable with mean 0 and variance, a vector $\bar{\beta}$ of size $n$ with $K = 40$ entries uniformly distributed on $\{1, \ldots, n\}$ and whose nonzero entries are iid Gaussian with mean 0 and variance 1 and $\bar{y} \triangleq \bar{X}\bar{\beta}$. For each problem, we carry out the following procedure. For each $m \in \{60, \ldots, 140\} \cap 2\mathbb{N}$, we let $X$ be the matrix from the first $m$ rows of $\bar{X}$, and $y$ be the first $m$ entries of $\bar{y}$. We then compute $\beta$ by minimising $f$ for this $X$ and $y$ using BFGS with 10 randomly generated starting points and declare "success" if $\|\beta - \bar{\beta}\|_2 \leqslant 10^{-3}$ for one of these starting points.