# OpenReview forum: "Smooth Bilevel Programming for Sparse Regularization"
_NeurIPS.cc/2021/Conference — NeurIPS 2021 Poster_

### Official Review · Reviewer_tRPK · 2021-06-29

**Rating:** 7
**Confidence:** 2

**Summary:**

The authors study regularized regression with a sparsity inducing constraint. (In the appendix also the extension to the low-rankness inducing case is considered.) The authors propose a new non-convex formulation, which is optimized via BFGS. The non-convex formulation is motivated by the well-known IRLS (Iteratively Reweighted Least Squares) algorithm.
Extensive numerical simulations are conducted on a wide variety of problems and the proposed algorithm is compared with a number of state-of-the-art approaches. While the proposed algorithm does not always perform best, it is in all cases at least competitive to the best performing approach.
Moreover, the authors prove that the induced non-convex function, which is optimized in their approach, is ridable -- meaning that all local minima are global minima and the Hessians of all saddle points have at least one negative eigenvalue.

**Main Review:**

In general, the paper is well-written and easy to read (although there are some typos, see below). The authors propose a new algorithm and demonstrate in experiments its versatility for a diverse set of problems. I clearly see the strengths of the paper on the empirical side. As the algorithm is new and appears to work well compared to a large class of existing algorithms on a wide set of problems (as it has been examined thoroughly in the paper), I would recommend acceptance.

I feel that the results on the theoretical side are a bit weaker. The authors cite Lee et al., which guarantees convergence from almost every starting point. However, it is known this convergence guarantee does not give any rates. (In particular, the convergence might take exponentially long in the accuracy.) Hence, it is not so clear what the impact is of the theoretical results I feel that this should be pointed out by the authors.
Of course, it might be interesting to have some bounds on the computational complexity of the presented algorithm (or on the convergence rate), but I feel that it might be difficult to derive those (especially in the very general setting as presented in this paper). Can one derive such bounds in the setting where for example $R (\beta) = \Vert \beta \Vert_{\ell_1} $? It could be interesting if the authors could comment on these issues in their paper.

Questions:
-In l. 238 the authors define Non-cvx-LBFGS. How is this algorithm different to Noncvx-Pro. Is it not exactly the same? What am I missing?
-The authors motivate their algorithm by IRLS. It is known that IRLS works especially well in the non-convex case, i.e. one consider the $\Vert \cdot \Vert_{\ell_p} $-norm with $p<1$. Have the authors compared their algorithm to the non-convex IRLS formulation?

Nitpicks:
- l. 11: minima->minimum
-l. 66 detail->detailed
-l. 180 approach(es)
-l. 185 global minimums->minima

**Time Spent Reviewing:**

3

---

> ### Comment · Reviewer_tRPK · 2021-08-22
> **Response**
>
> The authors have addressed all my questions sufficiently. I stick to my original assessment that this is a good paper which should be accepted.

---

### Official Review · Reviewer_eSTH · 2021-07-14

**Rating:** 7
**Confidence:** 3

**Summary:**

Authors proposed a re-parametrization of the variational form of usually sparsity enforcing penalties (including the Lasso and its variations).
The proposed variational re-parametrization leads to a non-convex but smooth bilevel optimization problem, that authors propose to solve using usual L-BFGS.
In addition to an extensive benchmark, authors show that the potential saddle points of the new minimized function are not "pathological".

**Main Review:**

The idea seems significantly new to me. Experiments are extensive and this work appears as a solid  and promising tool for the field of composite optimization. The general idea is very clear, however the paper could explain more the implementation details, in particular the bilevel approach.

- l21, the types of problems consider does not seem to include the (dual of the) SVM, is it possible to extend your work to also solve it?
- l80 you mentioned support pruning techniques, and works relying on working sets. Could your approach be combined with such techniques in order to gain speedups? This could be especially useful since the bilevel proposed approach, which requires solving a $\min(m,n)$ linear system.
- l220 You mention the conjugate gradient algorithm. Do you solve the linear system exactly? If not at which tolerance do you solve it? Do you use warm start on it? Does the algorithm explode in time in $n$ and $m$ are too large?
How "robust" is it to approximation? Because it is known that L-BFGS with approximate gradients can fail.
- l238 Non-cvx L-BFGS approach, what do you mean exactly by "solving (2) by L-BFGS", you consider the problem as a function of 2 variables $u$ and $v$ and apply L-BFGS on it? Could authors clarify?
- l240 it is written $\min_{u, v} ... ||X(u - v) - y||^2$, do you mean $\min_{u, v} ... ||X(u \odot v) - y||^2$ ?
- in order for the paper to be more pedagogical and instantly implementable by practitioners, I would recommend to detail the proposed algorithms on the Lasso in appendix, at least the computation of the gradient.


**Time Spent Reviewing:**

8

---

> ### Comment · Reviewer_eSTH · 2021-08-25
> **Comments after rebutal**
>
> I thank the authors for clarifications.
> I think the paper is good and should be accepted.

---

### Official Review · Reviewer_Nz5H · 2021-07-15

**Rating:** 6
**Confidence:** 3

**Summary:**

The authors propose a bilevel optimization approach for solving regularized empirical risk minimization. The considered minimization problem ($\cal{P}_{\lambda}$) refers to a linear model with parameters $\beta$ that are regularized in terms of a sparsity enforcing term $R(\beta)$. More specifically, the authors treat regularizers $R$ that can be recast in a so-called quadratic variational form. Therewith, a reparametrization of the original problem can be regarded as a bilevel program with smooth inner problem. Subsequent to the introduction, the authors provide a characterization of regularizers that have a quadratic variational form, followed by a theoretical analysis and numerical experiments.

**Limitations And Societal Impact:**

Yes

**Main Review:**

To the best of my knowledge, the described approach to reformulate and reparametrize the regularized objective is novel. Also, I find the presented idea interesting because the smoothed objective allows for applying efficient algorithms like L-BFGS. Indeed, the reported numerical results indicate that the method is competitive and especially robust with respect to the choice of the regularization parameter. As far as I can see, the manuscript is technically correct. And finally, the authors provide extensive supplementary material.

On the other hand, I think that the scope of the paper is limited to linear models and a few specific (although common) regularizers, and there exist a variety of solvers that can handle these kind of problems. Therefore, although I think that this is basically a good paper, I am not sure whether it will gain high significance.

**Time Spent Reviewing:**

3

---

### Official Review · Reviewer_j7tM · 2021-07-17

**Rating:** 8
**Confidence:** 3

**Summary:**

This paper proposes a reparametrization of common objective functions in the sparse estimation literature (for linear models). This allows rewriting LASSO, group LASSO, low-rank estimation as a smooth bilevel problem, and then applying fast methods for smooth problems such as L-BFGS. Critically, the gradient to the outer problem has a simple form to compute. The smooth problem is shown to have no spurious saddles, although it is nonconvex. The paper then shows impressive empirical results on multiple problems of interest stemming from this simple reparametrization.

**Main Review:**

The paper is well written and clear. It showcases impressive speedups thanks to the proposed reparametrization for a variety of problems. It takes an unusual route, by examining how a non-smooth convex problem can be written as a smooth, non-convex problem without spurious saddles.

The proposed method shows strong results, in particular ones which are robust across different problems. The proposed reparametrization is elegant and seems efficient. It is directly applicable to sparse linear models which are very much used in practice. It is easy to implement, since it relies on readily available smooth optimizers + linear solvers. The method would also directly benefit from improvement in these types of solvers.


**Time Spent Reviewing:**

2

---

### Author Response · Authors · 2021-08-05
**Response to Reviewer Nz5H**

“the scope of the paper is limited to linear models”

Response: it is possible to apply our methodology to non-linear models (such as neural networks) but of course in this case, the inner-minimization would not be solvable by a linear system, and this would require sub-iterations. This would make the method less attractive and difficult to analyze theoretically, but this might still be a valuable way to tackle sparsity enforcing regularizations. We will mention this in the paper, leaving this study for future work.

“there exist a variety of solvers that can handle these kind of problems”

Response: We believe that a strong point of our approach is that it works well for a large variety of settings (different regularizer, different type of matrix X and parameter lambda) which does not seem to be the case for SOTA methods.

---

### Author Response · Authors · 2021-08-05
**Response to Reviewer eSTH**

"l21, the types of problems consider does not seem to include the (dual of the) SVM, is it possible to extend your work to also solve it?"

Response: The dual SVM problem has a box constraint, so it is unclear how to apply the Non-Cvx VarPro method here. It is possible to add a l^1 penalty to the primal SVM, but the hinge loss makes the sub-problem not solvable in closed form using a linear system solver. This would necessitate to run sub-iterations, which makes the method less appealing (but this might still perform well). We will mention the possibility to use sub-iterations in the revised version (see also the response to Reviewer Nz5H).

"l80 you mentioned support pruning techniques, and works relying on working sets. Could your approach be combined with such techniques in order to gain speedups? This could be especially useful since the bilevel proposed approach, which requires solving a linear system."

Response: Yes, this is a very interesting venue for further investigations. The incorporation of support pruning would lead to linear systems of progressively smaller dimensions, leading to further speedups.

"l220 You mention the conjugate gradient algorithm. Do you solve the linear system exactly? If not at which tolerance do you solve it? Do you use warm start on it? Does the algorithm explode in time in and are too large? How "robust" is it to approximation? Because it is known that L-BFGS with approximate gradients can fail."

Response: Actually, in all our numerical experiments, to avoid these possible pitfalls, the linear system is solved exactly (using either a full or a sparse Cholesky solver). We will clarify this. For very large scale problems, one would indeed need to rely on an inexact solver, and controlling the impact of its approximation is an interesting avenue for future work (we will mention this).

"l238 Non-cvx L-BFGS approach, what do you mean exactly by "solving (2) by L-BFGS", you consider the problem as a function of 2 variables and apply L-BFGS on it? "

Response: Yes exactly, we consider the function to minimise as a function of u and v and apply L-BFGS to this (so L-BFGS is applied on twice the number of variables). We will clarify this in the final version.


"l240 it is written $\min_{u,v} ... ||X(u-v)-y||^2$, do you mean  $\min_{u,v} ... ||X(u\odot v)-y||^2$?"

Response: No, it is correct as written: this is a way in which L-BFGS was commonly applied to the Lasso in the literature, one splits the variable $\beta$ into its positive $u$ and negative parts $v$, so $\beta = u-v$. We are not sure what the earliest reference for this is, but it is for example mentioned in [Becker et al. 2019]. We will add a comment to ensure there is no confusion in the final version.

"in order for the paper to be more pedagogical and instantly implementable by practitioners, I would recommend to detail the proposed algorithms on the Lasso in appendix, at least the computation of the gradient."

Response: This is a good idea, we will write in the appendix a pseudo-code of the iterations for the Lasso (the associated formula for the gradient in the lasso and basis pursuit setting are given in Corollary 1) in the final version to make it clearer for practitioners.

---

### Author Response · Authors · 2021-08-05
**Response to Reviewer tRPK**

"I feel that the results on the theoretical side are a bit weaker. The authors cite Lee et al., which guarantees convergence from almost every starting point. However, it is known this convergence guarantee does not give any rates. (In particular, the convergence might take exponentially long in the accuracy.) Hence, it is not so clear what the impact is of the theoretical results I feel that this should be pointed out by the authors. Of course, it might be interesting to have some bounds on the computational complexity of the presented algorithm (or on the convergence rate), but I feel that it might be difficult to derive those (especially in the very general setting as presented in this paper). Can one derive such bounds in the setting where R=l1 for example ? "

Response: We agree with the reviewer. Even the l^1 case does not seem easy to analyze in terms of global rate. Of course,  local rates can be derived (e.g. for L-BFGS, there is local superlinear convergence). We will clarify and comment on this.

"Questions: -In l. 238 the authors define Non-cvx-LBFGS. How is this algorithm different to Noncvx-Pro. Is it not exactly the same? What am I missing?"

Response: Non-cvx-LBFGS applies L-BFGS to the full nonconvex formulation, that is, the function $G(u,v) := \frac12 h(v\odot v) + \frac12 \|u \|^2 + \frac{1}{\lambda} L(X(v\odot u),y).$  So, it is not the same as Noncvx-Pro. We will clarify this in the final version.

"-The authors motivate their algorithm by IRLS. It is known that IRLS works especially well in the non-convex case, i.e. one consider the lp-norm with p<1. Have the authors compared their algorithm to the non-convex IRLS formulation?"

Response: Doing an extensive study of the relative merits of IRLS vs Noncvx-Pro is indeed a very interesting avenue for future works. We did the following experiment: for compressed sensing examples (Random Gaussian matrices) and lp regularizer, IRLS is typically faster while for deconvolution examples, Noncvx-Pro is typically faster. For now, we did not perform extensive testing, so we decided not to include this, but we will mention this in the conclusion.

---

### Decision · Program_Chairs · 2021-09-27

**Decision:**

Accept (Poster)

**Comment:**

All reviewers found the reformulation interesting and appreciated the thorough experimental comparison. Remaining concerns include the (heavy) restriction to the least-square loss and the limited theoretical convergence guarantee. The authors' response acknowledged these issues and explained possible alleviations/extensions. Overall, there seems to be enough merit in the authors' approach and hopefully it could be further extended in the future.

Please incorporate the review, response and promised changes in the final revision. I would also suggest the authors refrain from using the term "bi-level programming," since the main theorem is simply a dualization of the loss, resulting in a standard min-max problem.